# Security Evaluation and Improvement of the Extended Protocol EIBsec for KNX/EIB

Tao Feng  and Bugang Zhang *

School of Computer and Communication, Lanzhou University of Technology, No. 36, Pengjiaping Road, Qilihe District, Lanzhou 730050, China; fengt@lut.edu.cn
* Correspondence: 212085400192@lut.edu.cn

**Abstract:** The European Installation Bus(EIB) protocol, also known as KNX/EIB, is widely used in building and home automation. An extension of the KNX/EIB protocol, EIBsec, is primarily designed to meet the requirements for data transmission security in distributed building automation systems. However, this protocol has some security issues in the request, key distribution, and identity authentication processes. This paper employs a formal analysis method that combines Colored Petri Net (CPN) theory with the Dolev-Yao attack model to evaluate and enhance the EIBsec protocol. It utilizes the CPN Tools to conduct CPN modeling analysis on the protocol and introduces a security assessment model to carry out intrusion detection and security assessment. Through this analysis, vulnerabilities in the protocol, such as tampering and replay attacks, are identified. To address these security concerns, we introduce hash verification and timestamp judgment methods into the original protocol to enhance its security. Subsequently, based on the improved protocol, we conduct CPN modeling and verify the security of the new scheme. Finally, through a comparison and analysis of the performance and security between the original protocol and the improved scheme, it is found that the improved scheme has higher security.

**Keywords:** EIBsec protocol; formal analysis; security evaluation; Dolev–Yao attack model; CPN

## 1. Introduction

In recent years, the security issues of Building Automation Systems [1] have become a hot research topic, among which the EIBsec protocol is used as the KNX data protection [2] protocol in building automation systems. KNX/EIB [3] bus technology can realize the control of heating, ventilation, air conditioning, lighting, and shading systems, so it is widely used in BAS. However, while it offers convenience, there are also many security risks between devices connected by the bus. In particular, when communicating between devices, due to the incomplete security mechanism of the protocol, user identity information is leaked and communication data are intercepted. Additionally, fire alarm systems, system intrusion and access control systems must resist not only random interference, but also deliberate attacks by third parties who want to disrupt their actions, like supply chain attacks, third-party plugin vulnerability attacks, Phishing, etc., so they have higher security requirements. For example, the BACnet/IP [4,5] protocol, which is also widely used in BAS, has authentication defects. The sender and receiver obtain the session key through the key server in this protocol, but there is no storage and management of historical keys. Therefore, there are forward and backward security flaws. In addition, in the LonTalk [6] protocol, the sender and receiver authenticate their identities through pre-shared keys, but the two parties do not conduct key negotiation in advance, and there is a risk of authentication defects and key leakage.

As an extension of the KNX/EIB protocol, EIBsec employs the advanced AES [7,8] symmetric encryption algorithm for secure data transmission across the bus. This enables various devices to communicate and collaborate effectively. Consequently, the EIBsec

protocol finds extensive use in smart home devices and industrial control equipment for building automation.

For instance, iconic landmarks like the Oriental Pearl Tower in Shanghai rely on the EIBsec protocol to orchestrate the nightly display of synchronized colorful lights. Similarly, numerous shopping malls and renowned hotels depend on the EIBsec protocol to ensure the normal operation and data security [9,10] of ventilation systems, lighting systems, elevators, and other equipment. These Internet of Things (IoT) devices gather environmental information through sensors [11], converting them into digital data to uniformly transmit the information to the Internet of Things gateway [12,13], and transmit it to the EIBsec system for processing. The processed data are then sent back to the sensors [14], allowing them to adjust IoT devices based on the feedback. Simultaneously, the results are transmitted to managers through the Internet, facilitating the redesign or improvement of IoT devices using the collected data. Moreover, some businesses need to collect substantial user data through industrial control systems, including sensitive personal information such as images, videos, and audio recordings. There are some security issues involved in the above-mentioned transmission process. To address these issues, the EIBsec protocol should initially desensitize sensitive information when collecting and responding to data in the end device. There are three entities in the above process, terminal sensor, IoT gateway and EIBsec system. The data generated during this process are managed uniformly by data security engineers. Residential users should only have access rights; data administrators should have access, add and modify permissions; and system administrators should have access, modify, add and delete permissions.

Furthermore, this process should rigorously verify device information integrated into the EIBsec system. Additionally, identity verification of device administrators in the cloud environment is essential. In actual applications, the security requirements are far greater than these. But in the actual environment, does the EIBsec protocol meet this series of security requirements? Administrators have repeatedly claimed that the EIBsec system is absolutely secure. However, in recent years, network attackers use diverse attack methods, such as Apache Log4j2 (CVE-2021-4101) remote code execution vulnerability exploitation, Shiro deserialization vulnerability exploitation (CVE-2019-12422), social engineering phishing, etc., which have caused a series of security problems, such as system host compromise, system permissions and data loss, and sensitive data exposure to occur frequently. These are enough to cause us to pay attention to the security of the underlying protocol of the system.

Therefore, this article focuses on the EIBsec protocol as its research subject. Using the CPN Tools [15] modeling tool combined with the Dolev–Yao attacker model, we analyze the message interaction process of the protocol, evaluate the security attributes of the protocol, identify potential security vulnerabilities [16], and ultimately put forth an improved plan. This plan aims to mitigate security risks and enhance the system's security attributes. The contributions of this article are as follows:

1. Apply CPN [17] theory and formal analysis methods to scrutinize the security vulnerabilities within the EIBsec protocol. Utilize the CPN Tools modeling tool for modeling and analyzing the original protocol, ensuring the model's consistency with the original protocol;

2. Based on the Dolev–Yao [18,19] attacker model, a security assessment model was integrated into the CPN model of the original protocol to perform intrusion detection and security assessment. This verification revealed the presence of replay and tampering attack vulnerabilities in the original protocol;

3. We introduce an improvement scheme to address the existing security vulnerabilities by incorporating hash verification and timestamp judgment into the original protocol, thereby enhancing its security. Subsequently, we utilize the CPN Tools [20] tool to create a CPN model based on the improved scheme. We then apply the same security assessment model to validate the improved scheme;

4.  Finally, we analyze and compare the performance and security of the original scheme and the improved scheme;

The structure of this article is as follows: In Section 2, we discuss related work and introduce the basic knowledge of the protocol. Section 3 uses the CPN Tools tool to model and analyze the EIBsec protocol and verifies the consistency between the CPN model and the original protocol based on the state space report. In Section 4, we introduce the security assessment model to conduct security assessment and discover the security vulnerabilities of the protocol. In Section 5, we provide an improvement method for the security vulnerabilities in the protocol and conduct CPN modeling based on the improved scheme. The same security assessment model is introduced into the improved CPN model to verify the security of the new scheme. In Section 6, the performance and security of the new scheme are mainly analyzed and compared. Section 7 summarizes the full paper and looks forward to future work.

We found that the new scheme can resist tampering attacks and replay attacks through modeling evaluation. By comparing the original scheme and the new scheme in Section 7, we found that the new scheme has higher security.

## 2. Related Work

The security of data transmission in building automation systems is extremely important for individual users or for a business unit. Scholars in the field of information security have proposed various solutions for the secure transmission of data in building automation systems in communication protocols. The authors of [21] proposed a dual-factor authentication scheme, claiming that it can provide forward encryption and resist sensor capture attacks, user tracking attacks, and DOS attacks. This scheme is logically safe, but the author did not use formal analysis tools for modeling and analysis. Moreover, this scheme requires a large amount of capital costs when invested in actual production environments, and the implementation risk factors are high. The authors of [22] propose a secure EIB protocol called SEIB, which uses 32-bit CRC verification to monitor unauthorized access and a 128-bit counter to prevent replay attacks. However, SEIB still has many security risks, such as unprotected communication tunnels and the use of weak encryption algorithms, which can easily crack the content. literature [23,24] use the KNX/EIB-based network simulation framework OMNeT++ [25] to build a model for the EIBsec protocol to test the performance of the protocol, but do not analyze the security of the protocol, nor establish a visual security assessment model. The authors of [26] pointed out the application characteristics of the EIBsec protocol in KNX/TP networks, but did not analyze the security flaws of the protocol.

In summary, there is currently no reliable solution to verify and improve the EIBsec protocol, and there is also a lack of a formal protocol modeling analysis method. Additionally, an attacker model is needed to perform security testing on the original protocol and the improved protocol. This article uses formal analysis methods to analyze the interaction process of the original protocol and uses the Dolev–Yao attacker model to conduct a security assessment of the original protocol. It is found that the protocol has a risk of man-in-the-middle attacks leading to the theft of session keys and user data. In view of the security flaws in the protocol, this paper proposes a new improvement scheme and conducts modeling analysis and security assessment of the improved new protocol. This article conducts formal modeling, analysis, and evaluation of the protocol based on CPN theory. Security testing is conducted by extracting key parts of the protocol to discover security flaws. After a security assessment, it was found that the protocol has replay attacks and tampering attack vulnerabilities initiated by man-in-the-middle attackers. This article will propose an improved scheme to solve the security vulnerabilities [2,23] of this protocol. In order to verify the effectiveness of the proposed scheme, a security evaluation of the new method was conducted. Finally, the security of the original scheme and the improved scheme was compared.

Compared with existing security research, the scheme proposed in this article has the advantage of formal verification. A new formal model-checking method was used to analyze the security of the protocol while ensuring consistency with the original model. In order to study the security vulnerabilities of this protocol, an attacker model is introduced. We also propose targeted improvement methods for the discovered security vulnerabilities, and conduct subsequent security verification of the new solution.

### 2.1. Tools Comparison

This article mainly focuses on the security mechanisms in the protocol, verifies the security of the EIBsec protocol, and identifies security vulnerabilities in the protocol. Therefore, it is assumed that the communication is reliable. Then, formal modeling tools are used to model and analyze the protocol. The current mainstream formal modeling tools include Scyther [27], Tamarin [28], ProVerif [29], and CPN Tools [20].

The Scyther tool can only verify the confidentiality of the protocol, and it is not very effective at verifying other properties such as availability and integrity. In addition, the ability of Scyther in protocol verification is relatively weak, and for complex protocols or protocols that require more sophisticated analysis techniques, it cannot fully cover all security vulnerabilities. Tamarin only supports a limited range of protocol models; for example, it does not support non-deterministic state transitions, complex data structures, and computations, among others. Therefore, some complex protocols may not receive full verification in Tamarin, and the models generated by Tamarin may become very large, requiring a significant amount of memory and computational resources. ProVerif is a highly abstract tool that is typically used to verify small-scale protocols. As the protocol becomes more complex, it may not be able to handle all cases correctly. The verification time of ProVerif is influenced by the size and complexity of the protocol, so verifying large-scale protocols may take a long time.

CPN Tools is a tool for modeling, simulating, and analyzing Colored Petri Nets. It supports various models, including static and dynamic models, and can be easily modified and extended. It has an intuitive user interface that helps users create, edit, and analyze CPN models. Users can directly edit Petri nets through the graphical interface, or use advanced features to define Petri nets. It supports multiple analysis techniques, including simulation analysis, performance analysis, model checking, and state space analysis, making it convenient for users to perform comprehensive analyses of models. It enables hierarchical modeling and analysis of complex protocols for the verification of security mechanisms. Users can define the necessary data types through the ML language to detect the protocol's confidentiality, integrity, and availability and identify potential vulnerabilities. However, CPN Tools uses a specialized Petri net drawing language (ML), so users need to have a certain Petri net foundation to use the tool for modeling and analysis. Therefore, the learning curve is steep and may require some time and effort to learn and become proficient with the tool.

Compared to the three automated protocol security verification tools mentioned above, CPN Tools has several advantages in the manual analysis of protocols. Additionally, the high degree of freedom in the CPN modeling process is also one of its strengths. The state space is entirely controlled by the modeler, and different modeling and analysis methods can be implemented for different protocols. This is why CPN Tools are usually more effective than automated protocol verification tools in protocol verification. Therefore, this paper uses the CPN Tools to simulate the protocol.

### 2.2. Simple Use of CPN Tools and Petri Net Theory

CPN tools is a visualization tool that integrates simulation, editing, and creation. It can dynamically and intuitively present the interaction process between various links between the message requester and the service responder. The tools included in CPN include auxiliary tools, creation tools, hierarchical tools, network tools, network tools, simulation

tools, state vector space tools, style tools, etc. The CPN Tools accessory tool used in this article is shown in Figure 1 :

- Create tool: Used to create a CPN model, in which circles represent places, used to store data types, split and merge protocol message flows, and rectangles represent transitions, indicating the occurrence and enabling of things.
- NET tool: Used to create a new page
- Simulation tool: Mainly used to detect possible errors in modeling details.
- Hierarchy tool: Used to break down more detailed protocol underlying models.
- State space tool: used to generate state space reports and analyze modeling results.
- Style tool: Used to distinguish different types of attacks.

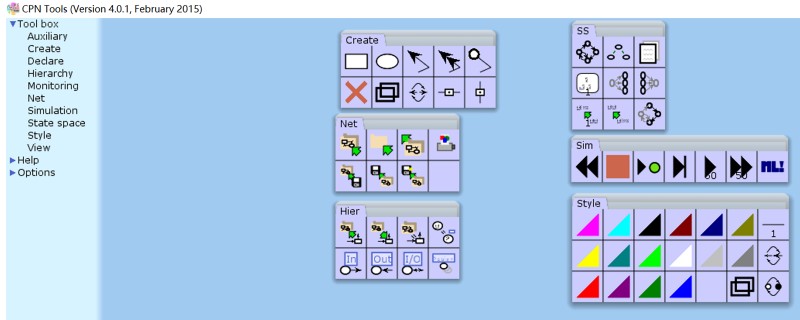

**Figure 1.** CPN Tools.

Let us introduce CPN with a simple visual example: In Figure 2, the red number 1 indicates the toolbox, the red number 2 indicates the function of the specific toolbar, the red number 3 indicates the model running steps and time, and the red number 4 is the area where variables and functions are defined in the CPN ML language.

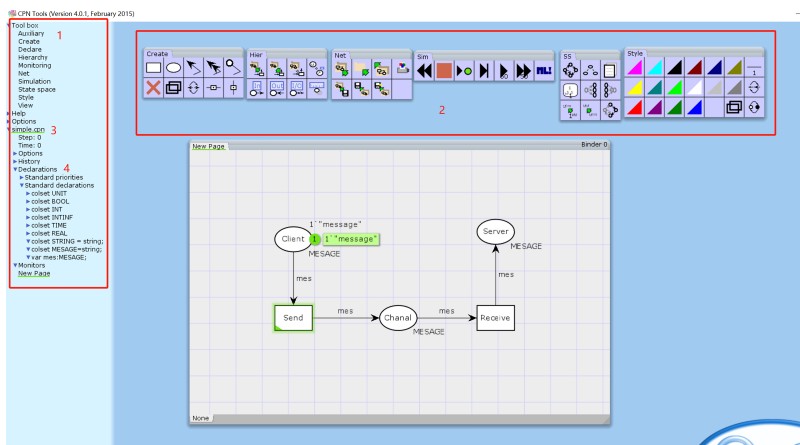

**Figure 2.** Simple model initial state of the protocol communication process.

In Figure 3, in the red number 1 area on the left, you can see that the client has sent MES type messages to the server. The red number 2 area is the generated state space report. Full represents the data that has run the entire process. Nodes and Arce in the State space and Scc graph are the same, indicating that there are no loops and the protocol runs completely. Only one of the Home markings and Dead Markings describes the correct modeling of the underlying protocol. The name of Home Markings is [3], indicating the status of the termination of the protocol operation. The correct modeling of the original protocol is very important for subsequent research.

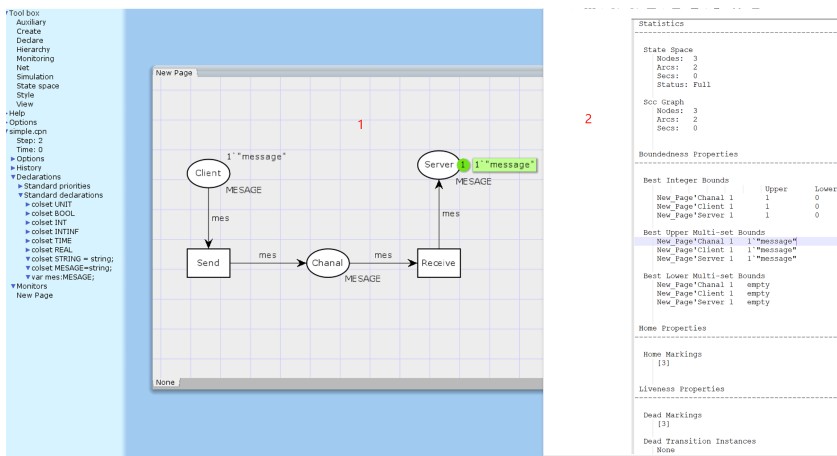

**Figure 3.** Simple model final state of the protocol communication process.

Petri net theory [30] is used in the CPN modeling process. It is a network that describes the relationship between events and conditions proposed by German scholar Carl Adam Petri in his doctoral thesis "Automata Communication" in 1962. This system model later became known as the Petri net. The term Petri net now refers to both this model and the theory developed based on this model. Petri nets are sometimes called network theory or Colored Petri Net theory. Petri nets are a formal method suitable for the specification and analysis of concurrent, asynchronous, and distributed software systems. Petri nets are divided into two types: location/migration Petri nets and advanced Petri nets. This article uses location/migration Petri nets. These nets are used in communication protocol verification, computer communication network performance evaluation and multimedia applications, software engineering, system reliability analysis, FMS modeling, analysis and control, System reliability analysis.

### 2.3. Preliminary Knowledge

As shown in Figure 4, The topology diagram of the EIBsec protocol is presented in a tree structure, mainly divided into three layers: backbone layer, mainline layer, and line layer. Each layer has an Advanced Coupler Unit (ACU) [31] at least, which is mainly used for distributing session keys or group keys. The backbone layer is connected to the internet through a gateway, the mainline layer is connected to the line layer through ACU, and the line layer implements specific data transmission services. ACU is similar to standard KNX/EIB lines [32] or backbone couplers [33], which allows ACU to be compatible with KNX/EIB network traffic and handle data requests from different network segments. Moreover, this tree structure is beneficial for avoiding the occurrence of single-point failures and can also help minimize the consequences of DoS attacks. For example, when an ACU detects a DoS attack on its network segment, it will be able to isolate the affected network segment and prevent attackers from accessing the rest of the network.

ACU mainly consists of two parts:

1. Coupling unit: Implement standard coupler function.
2. Key service unit: Implement the necessary functions of a key server (distributing and generating keys, revoking keys, and limiting key lifecycle).

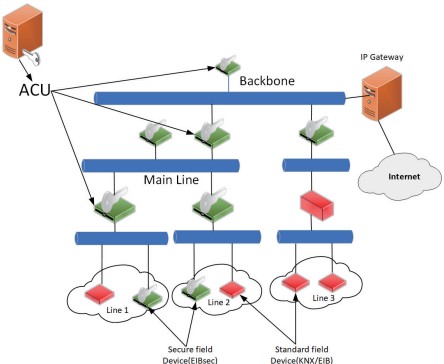

**Figure 4.** EIBsec topology.

## 3. Modeling for EIBsec Protocol

The EIBsec protocol is an extension of the KNX/EIB protocol, which provides protection for management communications and process data communications and is mainly used to protect data from malicious attacks during transmission. This protocol contains three communication entities during the communication process, namely: Entity A, Entity B, and ACU. In the EIBsec protocol, identity authentication is divided into two stages: the first stage is for the distribution of session keys, and the second stage is for identity authentication between entities. The protocol uses the AES-128 symmetric encryption algorithm to encrypt request and response messages in the key distribution phase and identity authentication phase. This article mainly studies the security flaws existing in the session establishment process, models and analyzes the protocol, and evaluates the security mechanism of the protocol. The flow chart and description of the symbols required in the modeling process are shown in Table 1.

**Table 1.** Symbols and Descriptions.

| Symbol | Description |
| --- | --- |
| Entity A/B | Participant A/B |
| ACU | Key distribution server |
| N1/N3 | generated by entity A |
| N2/N4 | generated by entity B |
| $N_1B$ | Verify response from entity B |
| $N_2A$ | Verify response from entity A |
| N3′ | (N3−1) |
| N4′ | (N4−1) |
| NA/NB | used to calculate the KA and KB |
| KA and KB | dynamic node keys |
| AddrA/AddrB | The address of Entity A/B |
| $K_{SB\_AB}$ | Session key or group key |
| Verify | Validating and comparison |
| Encr/Decr | Encr/Decr (k, message) |
| timestamp | Timestamp |
| ‖ | String concatenation |

The message flow diagram of the protocol is shown in Figure 5:

Phase 1: Key Distribution

Step 1: Entity A sends a session key request message to the corresponding key distribution server ACU, which includes random numbers N1, NA, and AddrA of Entity A, as well as the request address AddrB of Entity A.

Step 2: After receiving the session key request message, the corresponding ACU sends an initialization connection message to entity B based on the request content, which includes the address AddrA of entity A and the random number NB.

Step 3: Entity B responds to ACU with an encrypted message based on the connection request content, which contains a random number N2 and is encrypted using Entity B's key KB.

Step 4: After receiving the response message from entity B, ACU decrypts the response message and then responds to entity A and entity B with a message containing the session key. The message responding to entity A contains the session key $K_{SB\_AB}$ and Random number $N_1B$ of entity B, Using KA to encrypt messages containing $K_{SB\_AB}$ and $N_1B$. The message responding to entity B contains the session key $K_{SB\_AB}$ and Random number $N_2A$ of entity A, Using KA to encrypt messages containing $K_{SB\_AB}$ and $N_2A$. After this step is completed, it indicates that ACU has completed key distribution, followed by entity A and entity B for identity authentication.

Phase 2: Identity Authentication

Step 5: Entity A and entity B use KA and KB to decrypt the key response message and obtain the session key $K_{SB\_AB}$. At the same time, entity A obtains the random number $N_1B$ and entity B obtains the random number $N_2A$. Entity A initiates an identity authentication connection request to entity B. The request contains a new random number N3, using the session key $K_{SB\_AB}$ to encrypt the request message, and then sends the encrypted message A_Auth_Connect_Request to entity B.

Step 6: After receiving the request message A_Auth_Connect_Request, entity B uses the session key to decrypt and obtain the random number N3. At the same time, the random number N3' (N3$-$1) is calculated according to the agreed calculation rules. Then, we use $K_{SB\_AB}$ to encrypt N3' and the new random number N4 and send the encrypted message A_Auth_Connect_Reply to entity A.

Step 7: After receiving the A_Auth_Connect_Reply message from entity B, entity A decrypts and obtains the random number N4. Entity A uses the agreed calculation rules to recover N3 (N3' +1) and Compares N3 and N3'(N3' = N3 of original EntityA), if expectations are met, subsequent certification will be carried out. At the same time, Calculating N4' (N4$-$1) and encrypting messages containing the random number N4' using the session key $K_{SB\_AB}$ to form an A_Auth_Connect_Response message, it responds to entity B; entity B decrypts the message, obtains N4', then restores N4 (N4' + 1), and compares recovered N4 and N4' (N4' = N4 of original Etity B). If the expectations are met, the session between entity A and entity B is successfully established.

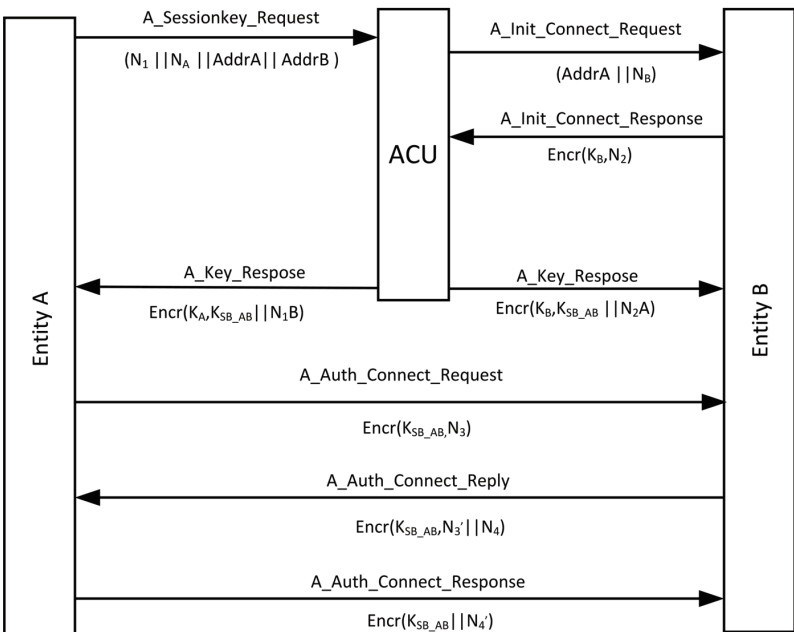

**Figure 5.** Session establishment in EIBsec.

### 3.1. Related Color Set Definitions

During the process of key distribution and identity authentication, protocols generate various types of messages, so before modeling, it is necessary to use CPN tools color set expressions to define the required data types. Table 2 is the definition of key color sets required in the modeling process. The table displays the expression of composite color sets, which are composed of a single color set based on the color set keywords in CPN tools. Defining the color set in advance is beneficial for establishing subsequent CPN models.

**Table 2.** The model's color sets the definition.

| Symbol | Meaning |
| --- | --- |
| NA1 | colset NA1 = product AddrA* Nonce; |
| KN | colset KN = product Nonce*Nonce; |
| MSGA | colset MSGA = record m:msgA*k:Nonce; |
| MSGB | colset MSGB = record m:msgB*k:Nonce; |
| AUTH1 | colset AUTH1 = product randomNumber*KAB; |
| AUTH2 | colset AUTH2 = record n:NN*k:KAB; |
| AUTH3 | colset AUTH3 = product randomNumber*KAB; |

### 3.2. Formal Modeling Process

In the previous section, we defined the relevant color set data types to lay the foundation for the modeling process. In the next section, we utilize the CPN Tools tool to model and analyze the protocol, aiming to identify security vulnerabilities within the protocol. Firstly, we assume that the protocol is secure and then proceed to create a model based on the message flow of the protocol. To streamline the complexity of the protocol, we employ a hierarchical modeling approach, dividing the CPN model of the protocol into top and bottom layers.

There are three communication entities in the CPN model: entity A, ACU, and entity B. This model captures static characteristics such as protocol status through the distribution of location tokens. Dynamic properties, including state changes, are described using rules and token flows that facilitate transitions. In the context of CPN modeling, taking the first message nnaa as an example, the EIBsec protocol can be formally defined using the following nine-tuple [34]:

$$EIBsec = (\Sigma, P, T, A, N, C, G, E, I) \tag{1}$$

$$Colorset \Sigma = closet Nonce = with NA|NB|NC|N1|N2|N3|N4|N3'|N4';$$

colset AddrA = string; colset AddrB = string;

- Place set P = AddrA, NA, AddrB, N1, config, S1;
- Transition set T = ASR, mes;
- Directed arc set A = AddrA→ASR; NA→ASR; AddrB→ASR;

  N1→ASR; ASR→config; config→mes; mes→S1;
- Node function N = AddrA→ASR: (AddrA, ASR); NA→ASR:

  (NA, ASR); AddrB→ASR: (AddrB, ASR); N1→ASR: (N1, ASR);
  ASR→config: (ASR, config); config→mes: (config, mes); mes→S1: (mes, S1);
- Color function C = AddrA:STRING; NA:WHIT; AddrA: STRING;

  N1:WHIT; config; NNAA; S1:NNAA
- Alert function G = NULL;
- Arc expression function E = addra→ASR:addra; NA→ASR:NA;

  addrb→ASR:addrb; N1→ASR:N1; ASR→config:addra, NA, addrb, N1;
  config→mes:config; mes→S1:mes;
- Initialization function I = addra:addra; NA:NA; addrb:addrb;

NA:NA; config, S1:NULL;

### *3.3. CPN Formal Modeling for EIBsec Protocol*

Figure 6 is a top-level CPN model built based on the message flow of the EIBsec protocol. This model generally describes the process of key distribution and identity authentication in the protocol. The top-level model consists of Entity A, which which initiates the request; ACU, which is responsible for key distribution, and Entity B. The entire interaction process of the protocol is visually simulated; the double matrix represents the existence of underlying substitution transitions, which respectively represent Entity A, ACU, and Entity B. A single matrix in an alternative transition represents a transition that is used to encrypt, decrypt, and calculate messages during a session, The ellipse represents a place which is used to store messages during the session construction process. The line segment with an arrow represents the message delivery direction. Its main function is to pass the message to the corresponding entity for processing. Overall, the top-level model contains a total of three transitions and eight message locations.

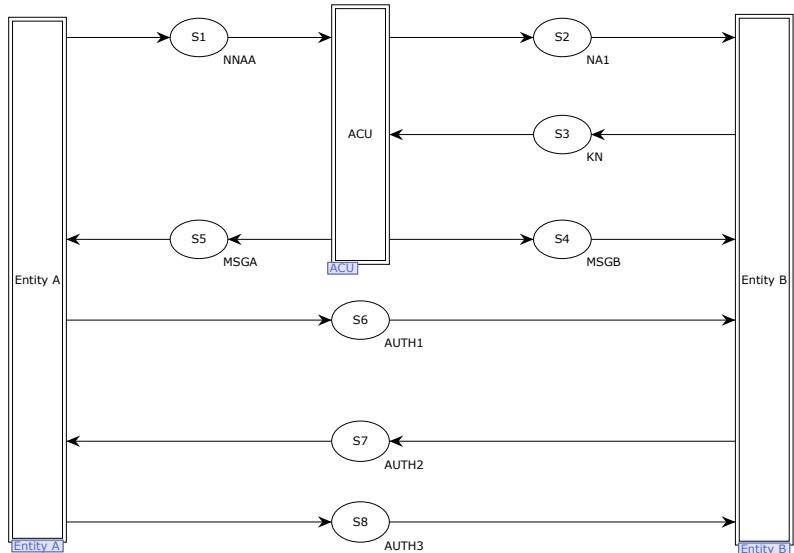

**Figure 6.** Top-level CPN mode of EIBsec protocol.

Figure 7 shows the internal CPN model of entity A's substitution transition. Firstly, Entity A sends a session key request nnaa message to the key distribution server ACU through the message store place S1. The message nnaa is composed of random numbers NA, N1, addra, and addrb. Place S5 receives the MSGA message sent by the key server ACU. The MSGA message is encrypted by ka, decrypted at the transition DmsgA to obtain the session key kab, and stored in the kab place. Transition Auth combines the session key kab and the random number N3 into an AUTH1 identity authentication request message, sends it to the S6 message storage place, and sends it to entity B through the S6 place.

The S7 place receives the identity authentication AUTH2 message from entity B. AUTH2 is composed of the composite random number nn and the session key kab. The composite random number nn is split through the transition Denn to obtain N3' calculated by entity B, and temporarily store N3' in the N3' place. Next, verification and calculation of N3' is carried out through transition verification to obtain the restored N3. Comparing N3' and N3 in transition comparison. If the expectations are met, the identity of entity B is verified and the random number N4 is output. The random number comes from entity B. Subsequently, the identity authentication of entity A is performed, and the random number n4 is subtracted by one through the transition convertN4 to obtain N4'.Transition encry uses the session key kab to encrypt the message containing the random number N4' to obtain the identity authentication message AUTH3, and sends AUTH3 to entity B through place S8 to verify the identity of entity A.

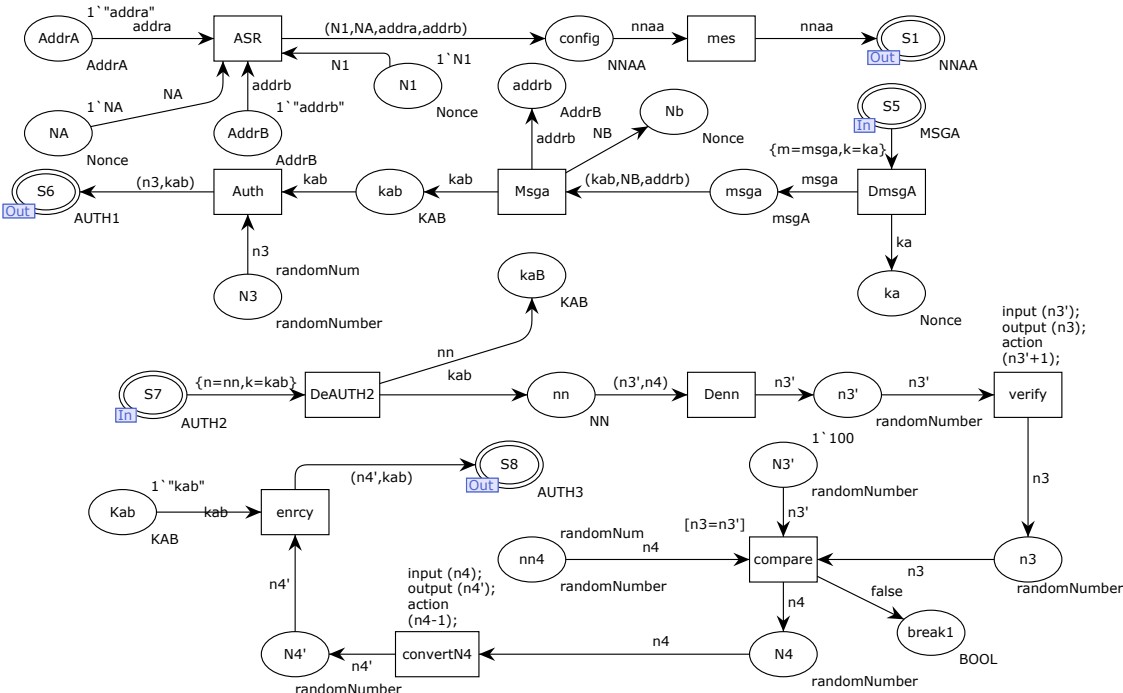

**Figure 7.** Internal CPN model of substitution transition Entity A.

Figure 8 shows the internal CPN model of the key distribution server ACU substitution transition. After receiving the request message from entity A through the S1 place, AUC splits the message NNAA according to the splitting principle and learns the entity object that entity A wants to establish a session connection. Subsequently, the ACU combines the request message (addra, NB) through the transition AICR. This message is composed of addra and the random number NB, and NB is generated by the ACU.

Place S3 receives the response message KN from entity B, which is a combination of kb and N2. At this time, ACU obtains the keys ka and kb generated by entity A and entity B, respectively. Finally, MSGA and MSGB are sent to Entity A and Entity B, respectively, through output places S4 and S5. Both messages contain the session key kab and carry random numbers NB and NA, respectively. At the same time, the messages are encrypted and transmitted using the keys ka and kb generated by each.

Figure 9 describes the internal CPN model of the substitution transition entity B. After receiving the request message sent by the ACU through S2, entity B learns the entity object that wants to establish a session connection. Entity B receives the MSGB message through S4, the transition DMSGB to use the key kb to decrypt, and it obtains the session key kab and the random number N2 generated by itself. If N2 is met as expected, it means that the session key is obtained by the expected key server distribution.

After entity B obtains the session key, it performs mutual identity authentication between entities. S6 receives the message encrypted by the session key and contains the random number n3. The transition auth uses the session key kab to decrypt the message. After the transition n3' obtains n3, it calculates n3' according to the action(n3-1) function. Transition N3N4 combines n3' and n4, and the combined message nn is encrypted using the session key kab through transition AUTH2 to obtain the AUTH2 message. At the same time, it is sent to entity A by S7. S8 receives the AUTH3 message, DeAUTH3 uses the session key to decrypt it, obtaining n4' and storing it in the n4' place, and transition verify verifies and calculates n4. The transition compare compares n4 and n4'. If they are equal, it proves that the identity of entity A is legal. Otherwise, the session establishment process ends, and the identity of entity A is not trustworthy.

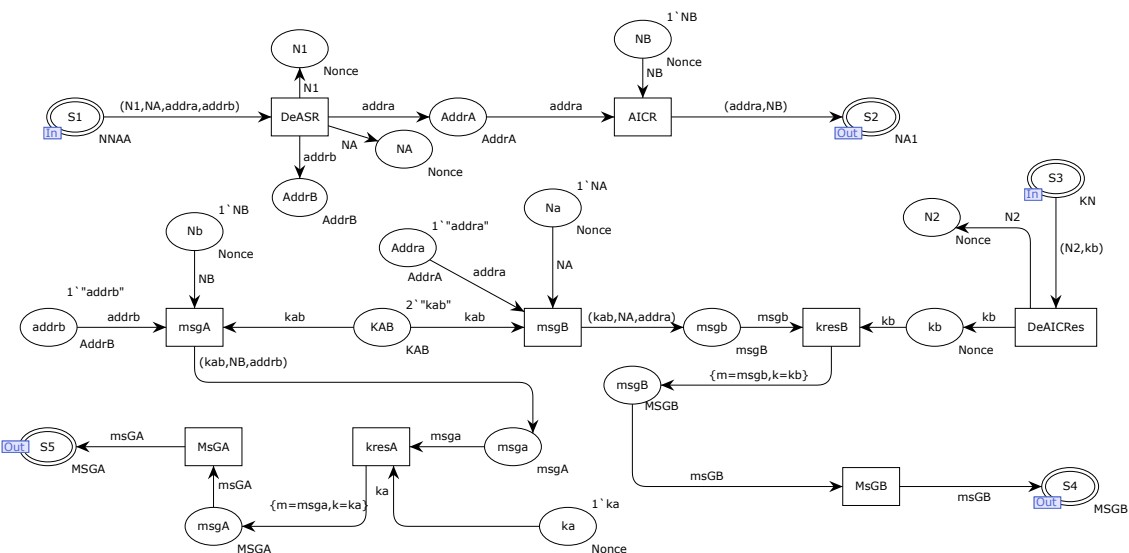

**Figure 8.** Internal CPN model of substitution transition ACU.

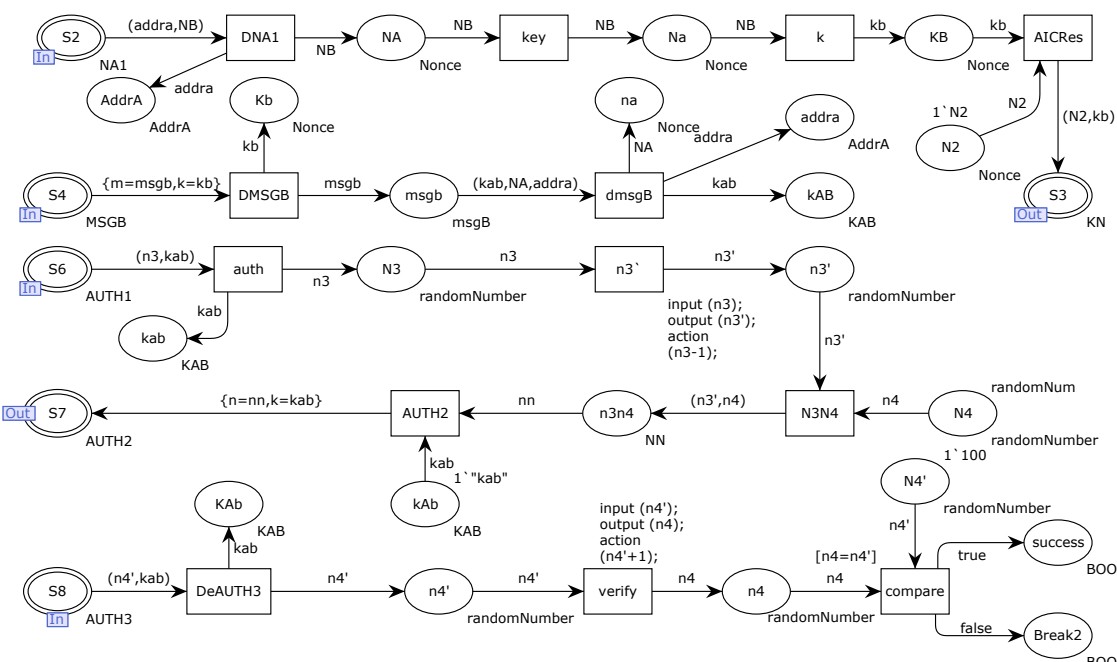

**Figure 9.** Internal CPN model of substitution transition Entity B.

### 3.4. Consistency Verification of the Original Protocol CPN Model

To verify the security properties of the EIBsec protocol, firstly, we must correctly model the protocol to ensure that the constructed CPN model is consistent with the original protocol. After the modeling using CPN tools is completed, we use the state space report generation tool to generate an experimental report and analyze whether the CPN model is consistent with the EIBsec protocol message from the experimental data. We expect that the CPN model has only one dead node. If this condition is met, it means that the constructed CPN model is correct.

According to the experimental data in Table 3, we found that the number of state space nodes and state space arcs is consistent with the number of Scc graph nodes and Scc graph arcs, respectively. It can be judged that the original EIBsec protocol CPN model does not have an infinite loop, the state space does not explode, and the model can run normally. The number of dead marks is 1 and the name number is 480, which means that

the last node after the model is completed is 480, which is what we expected, indicating the correctness of the original protocol CPN model. The dead transition is zero, which means that after the CPN model is completed, all transitions have been executed and there are no transitions that have not been executed.

**Table 3.** State space report of protocol original model.

| Type | Number | Name |
|---|---|---|
| State Space Nodes | 480 | / |
| State Space Arcs | 1096 | / |
| Scc graph Nodes | 480 | / |
| Scc graph Arcs | 1096 | / |
| Dead Marking | 1 | [480] |
| Dead transition instances | 0 | 0 |

## 4. Protocol Security Assessment Model

### 4.1. Dolev–Yao Attacker Model

The Dolev–Yao [35] model is designed based on the layered idea of the protocol from Dolev and Yao. It first considers whether there are vulnerabilities in the behavioral logic of the protocol itself, and then considers whether there are problems with the implementation method. This is exactly in line with the layered modeling method of CPN tools. This model assumes that the cryptographic system is perfect, that the attacker's knowledge and capabilities cannot be underestimated, and that the attacker can control the entire network. This assumption is considered from the perspective of the victim or the defender; that is to say, it prepares for the worst, assuming that the attacker can break into the protocol system. It makes us think about how to improve the protocol, thereby improving the security of the protocol system. As the defender or the victim, you should have this principle: never underestimate the knowledge and ability of the attacker.

Dolev and Yao also built an attacker model, describing in detail the behavior of the attacker:

1. The attacker can eavesdrop on network messages without being noticed by the main protocol;
2. The attacker can intercept and store messages in the network without being noticed by the host protocol;
3. An attacker can forge and send messages;
4. The attacker can participate in the operation of the protocol as a legitimate protocol participant.

In the Dolev–Yao threat model, attackers are almost omnipotent. It can be imagined that when each of us communicates on the network, we are communicating with attackers, and the messages we receive from the network are also sent to us by attackers. Therefore, our communication security and data security rely on cryptographic security protection.

### 4.2. EIBsec Protocol Security Assessment Model

A security assessment model of the EIBsec protocol is established based on the Dolev–Yao attacker model. This model includes three types of attacks initiated by attackers: tampering, replay, and spoofing. The red transitions and places in Figure 10 represent the tampering attack introduced in the underlying CPN model of the original protocol ACU. According to the message decomposition and combination rules in the attacker model, the attacker intercepts the message to be delivered to the ACU through A1 and stores it in the A1 place, and a tampering attack is launched through B1. Since in the original protocol, the request message initiated by entity A to establish a session is not encrypted, the message is passed to transition B2 through A3. B2 stores the split atomic messages in A4, A5, and A6, respectively. In transition B5, the random number NA is tampered with until it becomes NB, and the tampered message is reassembled to become the attack payload and stored in

MERGE. The attack payload is replayed to the output place S2 through the transmission paths ATTACK, nc1, and ATTAcK.

The purple part in Figure 10 indicates that a replay attack was introduced in the underlying CPN model of the original protocol ACU, and a replay attack was launched in the transition MsGA and MsGB during the session key distribution process respectively. The blue part in Figure 10 indicates that the spoofing attack was introduced in the underlying CPN model of the original protocol ACU, in which DeASR, AICR, DeAIRes, MsgA, and MsgB launched the spoofing attack.

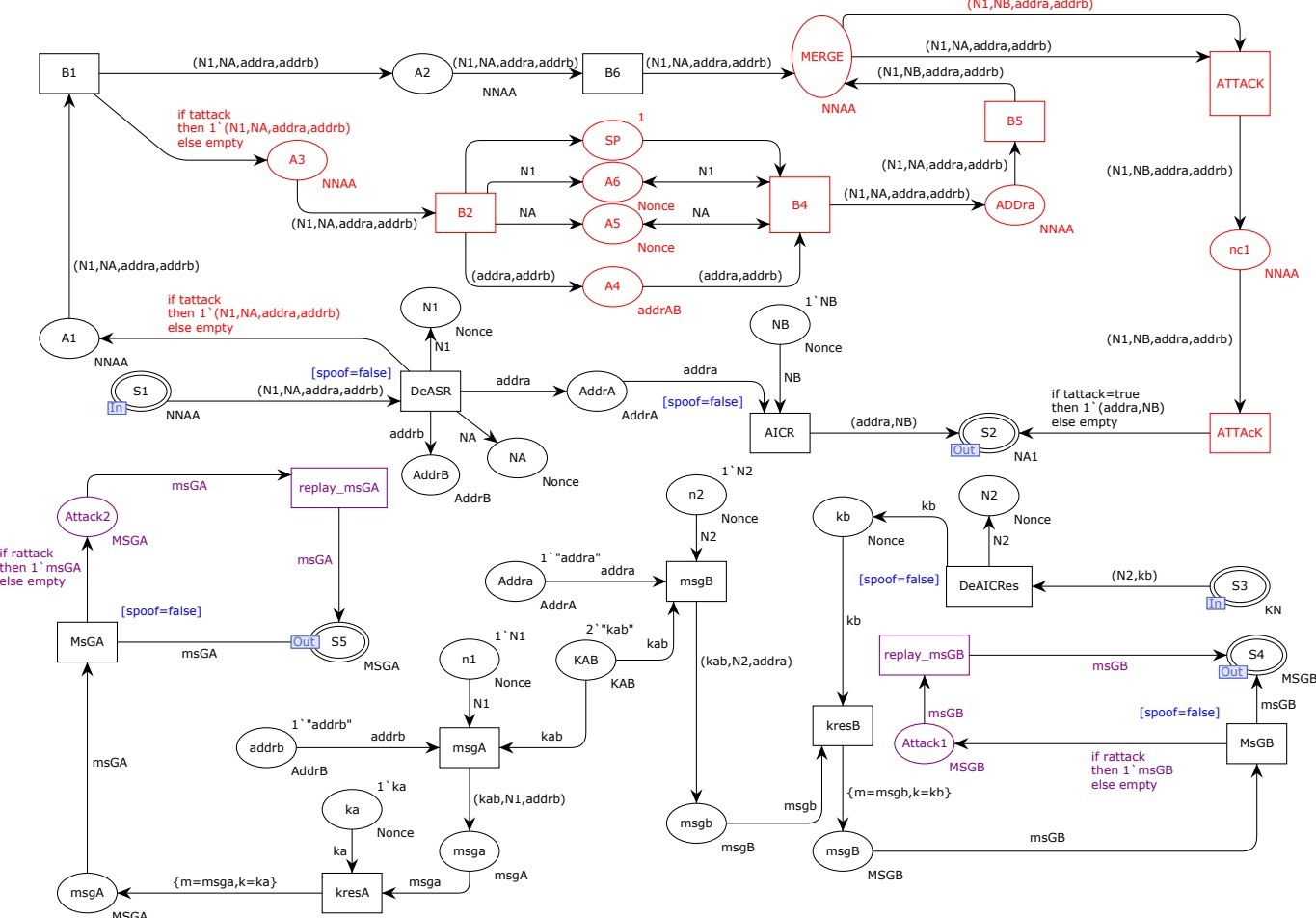

**Figure 10.** Security assessment model.

## *4.3. EIBsec Protocol Security Assessment*

In the previous section, we found security vulnerabilities in the original protocol based on the security assessment model. Table 4 is the state space report after introducing the attacker. We found that the number of nodes and arcs in the state space is consistent with the number of nodes and arcs in the Scc graph, indicating that each state node is fully reachable, the expression on the arc has been successfully executed, and the number of dead transitions is zero, indicating that the protocol security assessment model has been run in full. Among them, TAR-ATK represents a tampering attack, REY-ACK represents a replay attack, and SPF-ACK represents a spoofing attack.

According to the state space report in Table 4, we can find that three dead marks were generated after the introduction of tampering attacks, indicating that the original protocol has tampering attack vulnerabilities. When a session key request is initiated for the first time in the protocol, the request message is transmitted in clear text, which allows an attacker to tamper with the data in the request message at will after intercepting the message. In the security assessment model, we launch an attack by tampering with

random numbers, which causes the subsequent operation of the protocol to not return the expected results according to entity A's original request. The tampering attack destroys the confidentiality and integrity of the message. The replay attack in the security assessment model produced four dead marks, indicating that the original protocol had a replay attack, causing four nodes to fail. The attacker intercepts the message sent by the key distributor ACU at MsGA and MsGB and replays it to Entity A and Entity B many times. Replay attacks destroy the freshness of the message, causing the message to arrive late. The dead mark number of spoofing attacks in the model is one, indicating that there is no spoofing attack vulnerability in the original protocol.

As shown in Figure 11, the NodeDescriptor() function is used to query the status of 6520 transition and place nodes in the protocol running under a tampering attack, and the dead mark nodes generated due to the tampering attack can be queried. As can be seen from Figure 10, the tampering attack has been completed. The random number has been tampered with from NA to NB. The attacker successfully tampered with the protocol during the interaction process and passed the tampered message to entity B.

**Table 4.** Comparison of original protocol attacker model status space report.

| Type | TAR-ATK | REY-ATK | SPF-ATK |
|---|---|---|---|
| State Space Nodes | 6250 | 2805 | 480 |
| State Space Arcs | 21294 | 7997 | 1096 |
| Scc graph nodes | 6250 | 2805 | 480 |
| Scc graph Arcs | 21294 | 7997 | 1096 |
| Dead Markings | 3 | 4 | 1 |
| Dead transition | 0 | 0 | 0 |

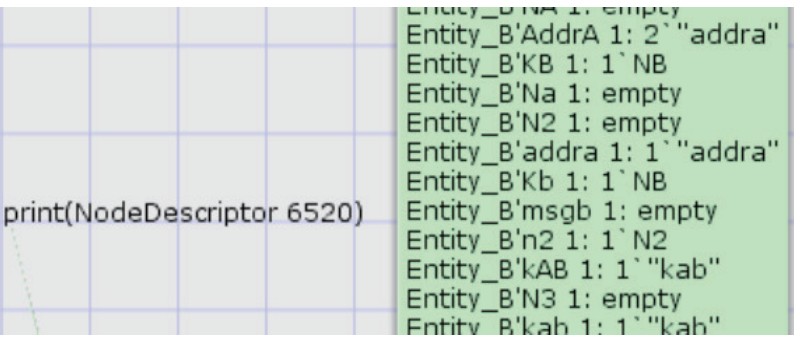

**Figure 11.** Querying the protocol termination status under tamper attack.

**5. New Scheme of EIBsec Protocol**

After passing the EIBsec protocol modeling evaluation in Section 4, it is found that there is no spoofing attack in the protocol itself. However, since the message is transmitted in plain text when sending the request and there is no time limit on the request and response of the message, there are tampering attacks and replay attack vulnerabilities in the original protocol. The improvement plan given in this article is: when entity A sends a request, it uses the parameters in the request message as the input of the hash function to calculate a hash value recorded as hash1, and sends the hash value together with the plaintext request message to the key distribution Server ACU. When ACU receives the request message, it uses the same hash function, uses the parameters in the request message to enter the hash function, and then calculates a new hash value recorded as hash2. Hash2 is compared with hash1 in the request message. In addition, considering the operation delay of transition in the model and the maximum time for normal message transmission, we added a timestamp based on the original protocol, which is in line with the idea of our design plan. In the CPN model, the timestamp is carried out at each stage of the message interaction process between entity A, ACU, and entity B, and a time upper limit threshold is set.

If the attacker tampers with the message, the generated hash value is different from hash1 in the original message, then the hash comparison fails in the ACU, the key request message is discarded, and the session establishment is interrupted. If the attacker simply intercepts the message and then replays it, the process will generate more time overhead, and the timestamp will exceed the upper time threshold. This chapter will conduct CPN modeling of the new scheme and use the same security assessment model to verify the effectiveness of the new scheme.

### 5.1. Protocol Improvement Scheme

Figure 12 is the message flow chart of the improvement scheme. The entire message flow process in the new scheme is also divided into two stages: distribution of session keys and identity authentication between communicating entities.

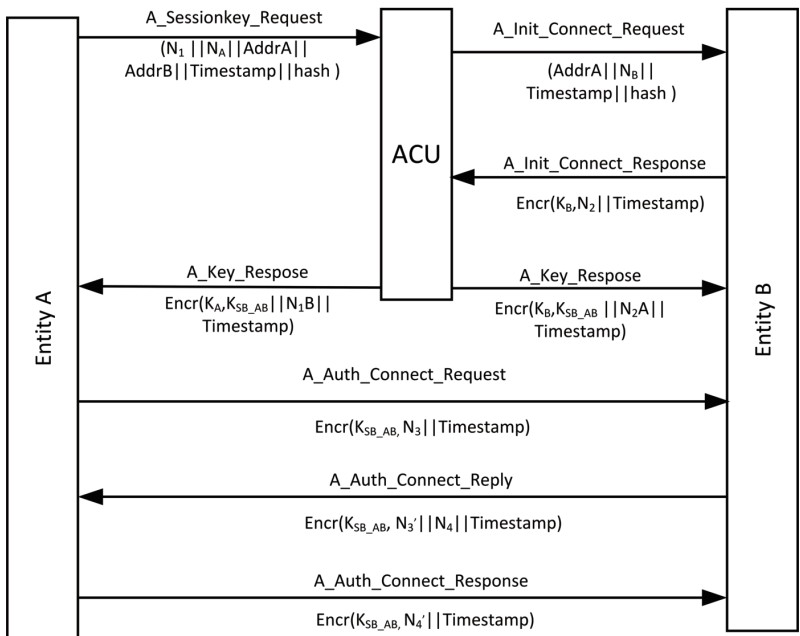

**Figure 12.** Model of the new scheme message flow.

The message interaction process of the improved new scheme protocol is as follows: Phase One: Key Distribution

Step 1: Entity A initiates a session key request message to the corresponding key distribution server ACU. The message contains the random number N1, NA, AddrA, and the address AddrB of the entity object that entity A wants to establish a session connection. At the same time, the data in the request message are used as a parameter to calculate a hash value recorded as hash1, the hash1 value is carried in the request message, and a timestamp is added when sending the message.

Step 2: After the corresponding ACU receives the session key request message, it obtains hash1 from the request message and uses the same method to calculate a new hash value recorded as hash2. Hash1 is compared with hash2. If the hash values are the same, the operation is followed up; otherwise, entity A's request is rejected. At the same time, ACU sets the timestamp threshold to determine whether the time spent by entity A in sending the request message has timed out.

Step 3: After entity B receives the instruction from ACU, it obtains the request parameters, uses the parameters in the request message to calculate a hash value, and then compares it with the hash value in the instruction. If the hash values are the same, it sends an encrypted message to ACU. Otherwise, the instruction from the ACU is discarded. At the same time, the timestamp threshold is set, and it is verified whether the timestamp times out and include the timestamp when sending messages to the ACU.

Step 4: After receiving the message sent by entity B, ACU decrypts the message, and then responds to entity A and entity B each with a message containing the session key. The message responding to entity A contains the session key $K_{SB\_AB}$, entity B's random number $N_1B$, and timestamp. KA is used to encrypt the message containing $K_{SB\_AB}$ and $N_1B$. The message responding to entity B contains the session key $K_{SB\_AB}$, entity A's random number $N_2A$, and timestamp. KB is used to encrypt the message containing $K_{SB\_AB}$ and $N_2A$. After this step is completed, it means that ACU completes key distribution, and then entity A and entity B perform identity authentication.

Phase 2: Identity Authentication

Step 5: Entity A and Entity B use KA and KB, respectively, to decrypt the key response message from the ACU and obtain the session key $K_{SB\_AB}$. At the same time, entity A obtains the random number $N_1B$, and entity B obtains the random number $N_2A$. Entity A and Entity B verify the time threshold. Entity A initiates an identity authentication connection request to entity B. The request contains a new random number N3, uses the session key $K_{SB\_AB}$ to encrypt the request message, timestamps it, and then sends it to entity B.

Step 6: Entity B receives the request message, uses the session key to decrypt it, and obtains the random number N3 and timestamp information. At the same time, the random number N3' is calculated through the action() function, and the timestamp is verified. Then, N3' is combined with the new random number N4 to form an A_Auth_Connect_Reply message, which is encrypted using $K_{SB\_AB}$, timestamp information is added, and the encrypted message with the timestamp is sent to entity A.

Step 7: After receiving the response message from entity B, entity A decrypts and obtains random numbers N3' and N4. Entity A uses the action() function to restore N3 (N3' + 1) and compares it with the N3 carried when requesting from entity A. If the two are equal, subsequent authentication will be performed. At the same time, N4' is calculated, and the message containing N4' is encrypted using the session key $K_{SB\_AB}$, and the timestamp information is carried together to form an A_Auth_Connect_Response message, which is responded to Entity B. After entity B receives the message, it decrypts the message and obtains N4', then restores N4 according to the agreed calculation rules, compares N4 and N4' (N4 = N4') and verifies whether the timestamp exceeds the set time threshold. If it meets As expected, the session between Entity A and Entity B is successfully established.

*5.2. New Scheme Model of EIBsec Protocol*

The improved EIBsec protocol also consists of three parts: entity A, ACU, and entity B. The new scheme adds hash verification and timestamp judgment into the original protocol to strengthen the security mechanism of the protocol. In this section, we use the CPN Tools tool to model and analyze the new scheme. In Section 5.3, we use the same security assessment model to verify the security of our new and improved scheme. The new scheme CPN model is mainly divided into the top-level model and bottom-level model. The top-level model is consistent with the top-level model of the original protocol, so this section only shows the underlying CPN model. The underlying model describes the key distribution and identity authentication process of the protocol message flow in the new scheme in more detail.

The first CPN mathematical model expression of the modified original protocol:

$$EIBsec = (\Sigma, P, T, A, N, C, G, E, I) \tag{2}$$

$$Colorset \Sigma = closet Nonce = with NA|NB|NC|N1|N2|N3|N4|N3'|N4';$$

colset AddrA = string; colset AddrB = string; closet T = int; colset HASH = string; val prdelay = 3; val trdelay = 6;

- Place set P = AddrA, NA, AddrB, N1, config, S1, Initial time, hash;
- Transition set T = ASR, mes, Hnnaat;

- Directed arc set A = AddrA→ASR; NA→ASR; AddrB→ASR;N1→ASR;

  Initial time→ASR; HASH→ASR; ASR→config; config→mes;
  mes→hnnaat; hnnaat→Hnnaat; Hnnaat→S1

- Node function N = AddrA→ASR:(AddrA, ASR); NA→ASR:(NA, ASR);

  AddrB→ASR:(AddrB,ASR); N1→ASR:(N1,ASR); t→ASR:(t,ASR);
  hash1→ASR:(hash1,ASR); ASR→config:(ASR,config);
  config→mes:(config, mes); mes→hnnaat:(mes, hnnaat);
  hnnaat→Hnnaat:(hnnaat, Hnnaat); Hnnaat→S1: (Hnnaat, S1);

- Color function C = AddrA:STRING; NA:WHIT; AddrA:STRING;

  N1:WHIT; config; NNAA;
  S1:NNAA; T:int; HASH:STRING; prdelay, trdelay:VAL;

- Alert function G = NULL;
- Arc expression function E = addra→ASR:addra; NA→ ASR:NA;

  addrb→ASR:addrb; N1→ ASR:N1; t→ASR:t; hash1→ASR:hash1;
  ASR→config:addra, NA, addrb, N1, t + prdelay + trdelay, hash1;
  config→mes:config; mes→hnnaat:cmes;
  hnnaat→Hnnaat:hnnaat; Hnnaat→S1:Hnnaat;

- Initialization function I = addra:addra; NA:NA; addrb:addrb; NA:NA; t:0;

  hash1:hash1; config, hnnaat, S1:NULL;

Figure 13 shows the underlying CPN model of the improved entity A. In the original protocol, we introduced timestamp and hash. The initial time of the timestamp is 0, and the hash value hash1 is obtained through the hash function. In practical applications, SHA-1 and MD5 can be selected to calculate hash values using the hash function. Place S1 sends message hNNaat to ACU. The message hNNaat is composed of (hash1, N1, NA, addra, addrb, t). Among them, hash1 is used to ensure the integrity of the message, and timestamp is used to ensure the freshness of the message. Place S5 receives the message {k = ka, m = msga} from the ACU, and uses [t <= 9] at the transition Msga to verify whether the timestamp has timed out. If the timestamp has not timed out, it means that the message has not been tampered with or repeated. The distribution of session key kab is completed. Place S6 sends a message ((kab, n3), t) to entity B. The session key kab encrypts the content to be sent and the random number n3 and adds a new timestamp to the encrypted message. Place S7 receives the message {k = kab, n = nn} from entity B, decrypts it at transition DeAUTH2, and verifies whether the message has timed out through the preset timestamp threshold of [t <= 45] at transition Denn. Message ((kab, n4′), t) is sent to entity B through place S8. n4′ = n4-1 in the message is calculated through the action(n4-1) function in the transition convertN4, using the session key kab, the message content and random number N4′ are encrypted, and a new timestamp is added to the encrypted message. In the new scheme, each message will carry a timestamp.

Figure 14 depicts the underlying CPN model of the improved ACU. ACU receives the hNNaat message through the input place S1, decomposes it at the transition DeASR, and obtains the request message parameters. Using (N1, NA, addra, addrb) as the input of the hash function to recalculate the hash value at the transition CalculateHash and storing the calculation result in hash2, the hash value is calculated at the transition compareHash. If hash1 is not equal to hash2, it means that the message has been tampered with. At this time, the key distribution server ACU discards the request message packet and interrupts the session establishment. On the contrary, place S2 will send the message (hash1, NB, addra, t) to entity B. Place S3 receives the response message (N2, kb, t) from entity B, and ACU verifies whether the timestamp message times out at the transition DeAICRes. Subsequently, the key distribution server ACU sends messages msGA and msGB to entity A and entity B, respectively. Both messages contain the session key communicated by entity A and entity B and the random number of the other party, and the message package carries a timestamp.

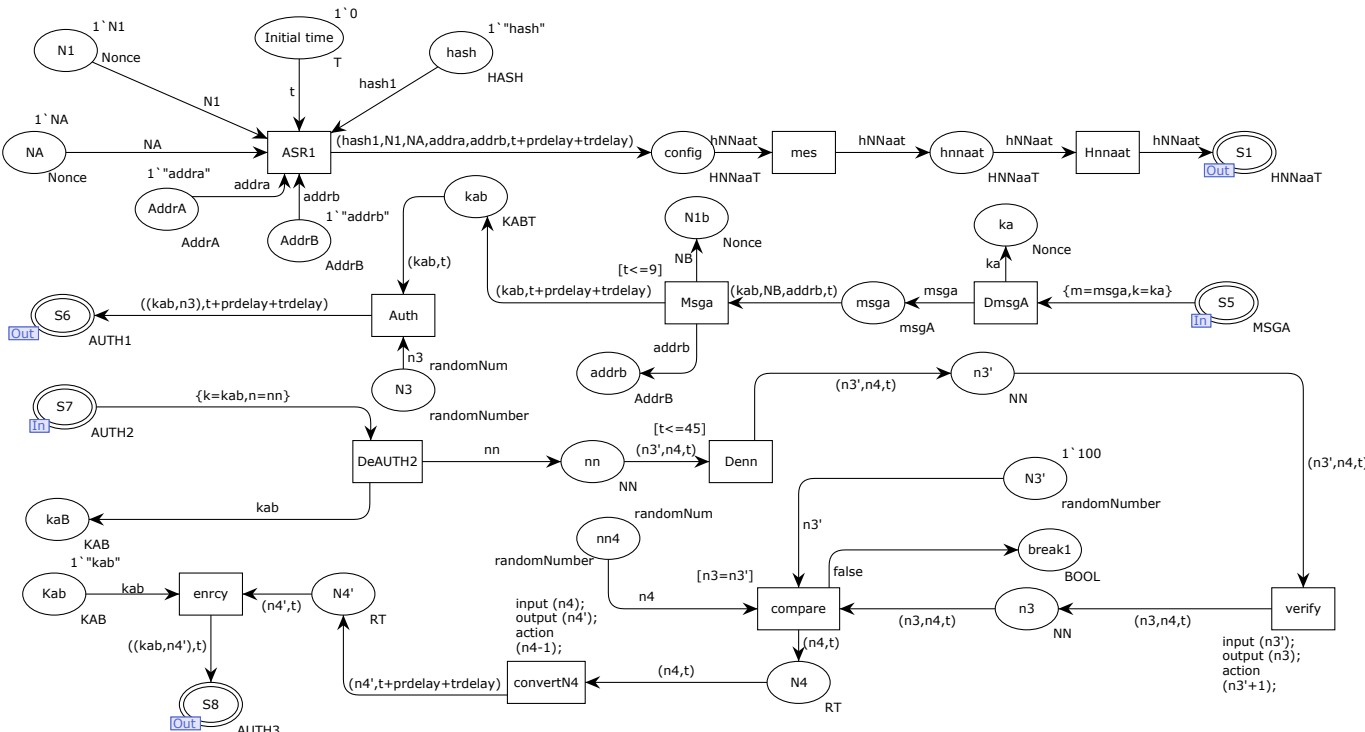

**Figure 13.** Internal CPN model of improved Entity A.

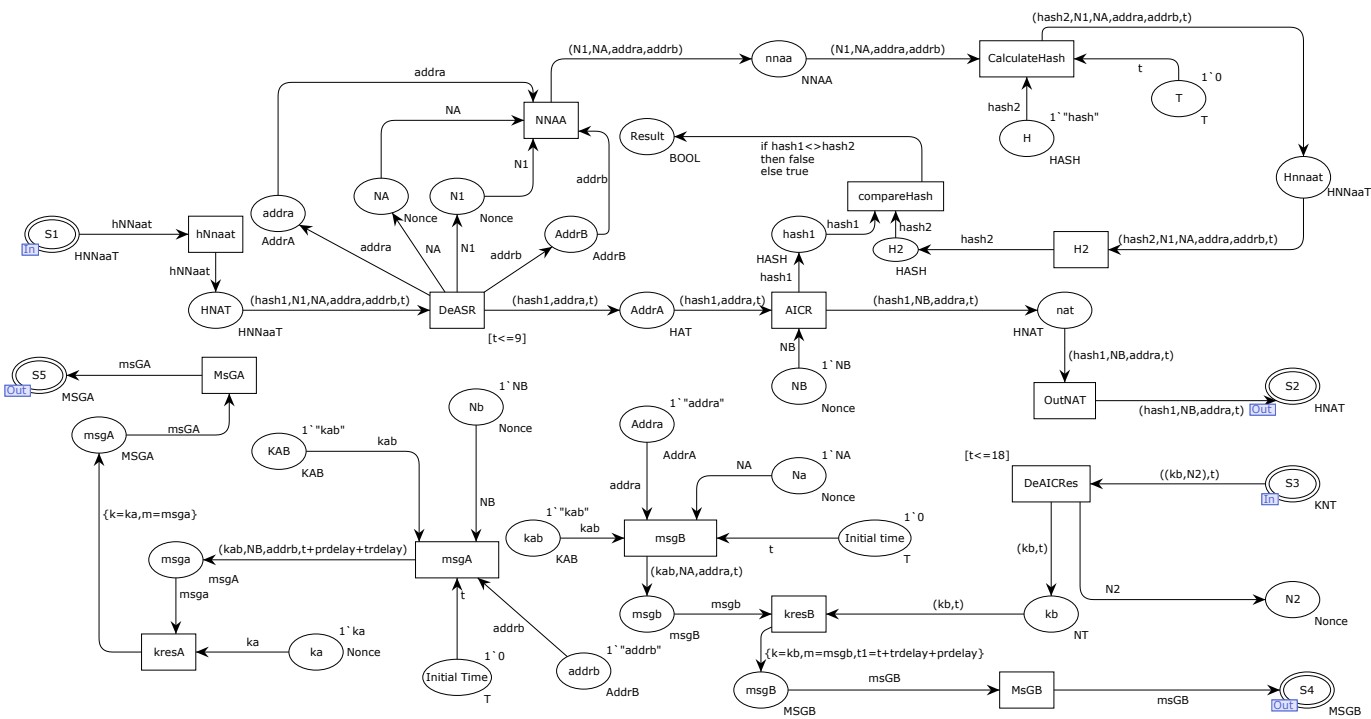

**Figure 14.** Internal CPN model of improved ACU.

Figure 15 describes the underlying CPN model of the improved entity B. The input place S2 receives the message (hash1, NB, addra, t), verifies the timestamp information at the transition DNA1, transition CalculateHash to recalculate the hash, and stores the calculation result in the variable hash3. Transition CompareHASH compares hash1 and hash3 and output place S3 to send a message ((kb, N2), t) to ACU. The input place S4 receives the {m = msgb, k = kb, t1 = t} message, verifies the timestamp information at the

transition DMSGB, and entity B obtains the session key kab in the message msgb. Place S6 receives the identity authentication request message ((kab, n3), t) from entity A. Using the session key kab previously distributed by ACU in the transition DeAuth, it decrypts the authentication message and verifies the timestamp. Place S7 sends {k = kab, n = nn} message to entity A. In the message, nn = (n3', n4, t + prdelay + trdelay), n3' = action(n3-1), and kab is used to Encrypt the message. Finally, it is sent with a timestamp to entity A. Place S8 receives the message ((kab, n4'), t), verifies whether the timestamp has timed out according to the upper time threshold [t <= 54] at transition DeAUTH3, and decrypts it to obtain n4'. At the transition verify, n4 is restored according to n4 = action(n4' + 1). At the transition compareN4, according to [n4 = n4'], it is judged whether the random number has been tampered with. If they are equal, it means that the session between Entity A and Entity B has been successfully established.

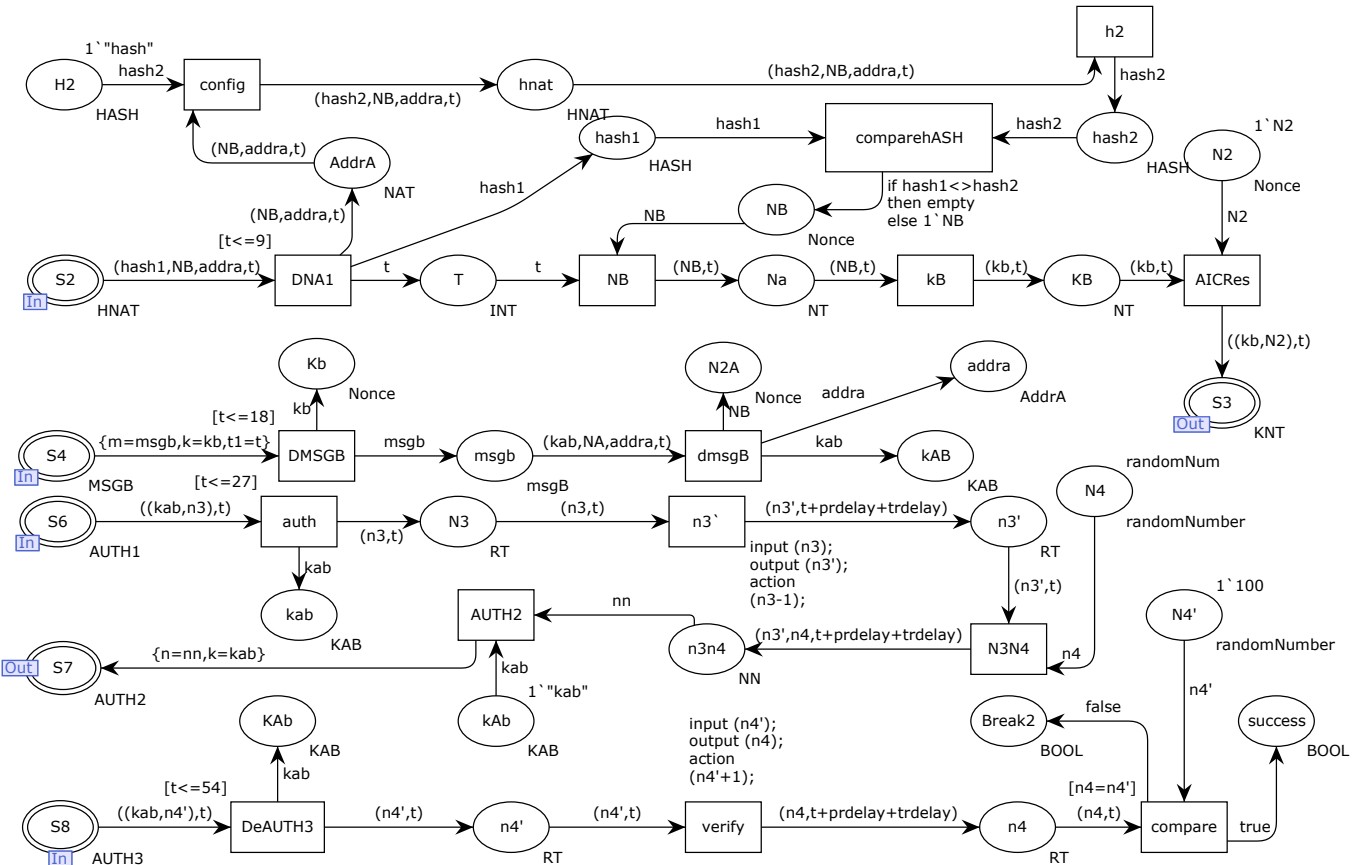

**Figure 15.** Internal CPN Model of Improved Entity B.

### 5.3. New Scheme Security Assessment Model

In Section 5.2 we strengthen the security mechanism of the protocol by adding times-tamps and hashes to the CPN model of the original protocol. As shown in Figure 16, in this section, we will introduce the same Dolev–Yao adversary evaluation model into the improved new scheme CPN model to attack our improved model. The evaluation model still includes three types of man-in-the-middle attacks: tampering attacks, replay attacks, and spoofing attacks. The attacker launches the attack at the key distribution server ACU, as shown in Figure 11. The red part represents a tampering attack, the purple part represents a replay attack, and the blue part represents a spoofing attack.

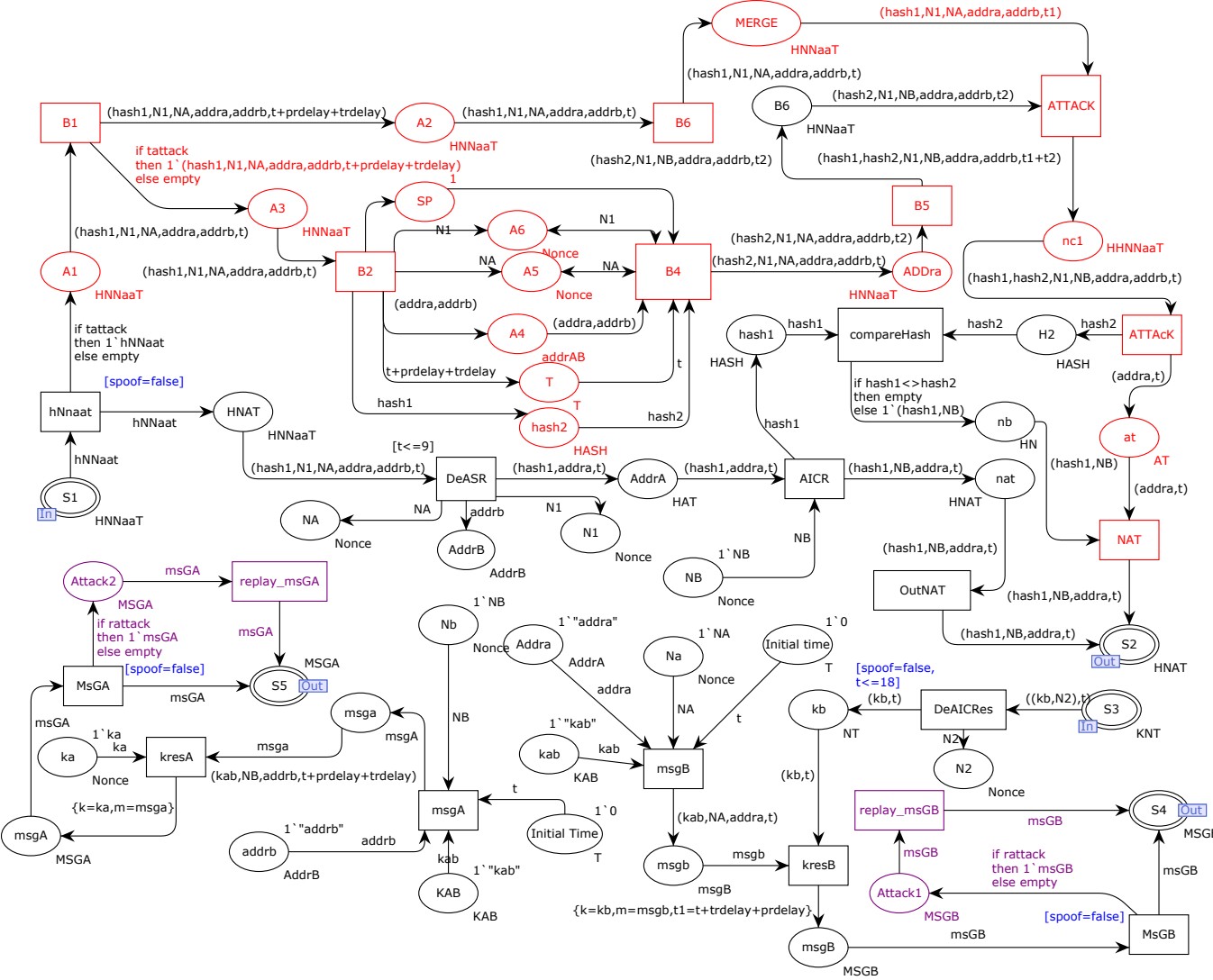

**Figure 16.** New scheme security assessment model.

*5.4. Security Evaluation of Improved Scheme*

In the previous section, we introduced three man-in-the-middle [36] attacks at the bottom of ACU to verify whether the improved scheme can effectively defend against the three attacks. In this section, we analyze the state space reports generated by these three attacks.

As shown in Table 5, by comparing the state space under three attack methods before and after the improvement, we can find that the state space nodes and state space arcs under the three attacks have increased significantly in the new scheme. This is due to the addition of a security defense mechanism in the model, making the model more complex. The number of dead nodes is reduced from 3 to 1 in the tampering attack. We can find that in the tampering attack, although the attacker modifies the random number. However, since the output place S2 needs to compare and verify the hash at the transition comparehash before sending the message, the tampered random number cannot pass the hash verification. Finally, the attacker cannot launch an effective tampering attack. The number of dead nodes is reduced from 4 to 1 in the replay attack. It can be found that after the timestamp is introduced, the attacker launching a replay attack will cause the message to time out, making the replay attack impossible. The dead nodes before and after the spoofing attack improvement are all one, illustrating there is no spoofing attack in the original protocol.

**Table 5.** Comparison of state spaces under three attack models.

| Type | Before Improvement | | | New Scheme | | |
| --- | --- | --- | --- | --- | --- | --- |
| | **TAR-ATK** | **REY-ATK** | **SPF-ATK** | **TAR-ATK** | **REY-ATK** | **SPF-ATK** |
| State Space Nodes | 6520 | 2805 | 480 | 8960 | 15,555 | 2760 |
| State Space Arcs | 21,294 | 7997 | 1096 | 30,072 | 56,332 | 8402 |
| Scc graph nodes | 6520 | 2805 | 480 | 8960 | 15,555 | 2760 |
| Scc graph Arcs | 21,294 | 7997 | 1096 | 30,072 | 56,332 | 8402 |
| Dead Markings | 3 | 4 | 1 | 1 | 1 | 1 |
| Dead Transition | 0 | 0 | 0 | 0 | 0 | 0 |

As shown in Table 6, by comparing the state space generated by the security assessment model before and after the improvement, it is found that the number of nodes and arcs in the improved state space has increased significantly, which is due to the addition of hash comparison and timestamp verification. The number of dead nodes has been reduced from 16 to 1, indicating that the improved scheme is effective and can effectively defend against tampering and replay attacks. Regardless of whether it is a legitimate user or an illegal attacker, the request message initiated must pass hash verification and timestamp judgment, and the message that cannot pass will be discarded. The security mechanism of the new scheme can effectively ensure that the data are not tampered with and replayed during the transmission process, ensuring the integrity, freshness, and confidentiality of the data during the transmission process.

**Table 6.** State space comparison of security assessment model

| Type | Before Improvement | New Model |
| --- | --- | --- |
| State Space Nodes | 42,245 | 50,320 |
| State Space Arcs | 162,968 | 197,323 |
| Scc graph Nodes | 42,245 | 50,320 |
| Scc graph Arcs | 162,968 | 197,323 |
| Dead Marking | 16 | 1 |
| Dead transition | 0 | 0 |

## 6. Performance and Security Analysis of New Scheme

### 6.1. Performance Analysis

The EIBsec protocol uses a symmetric encryption algorithm, and the time spent on running the protocol is mainly taken during the encryption, decryption, and verification steps. The time consumed includes generating random numbers NA, N1, N2, N3, N4, NB and addresses addra, addrb, represented by the symbol Tgeb; encryption, decryption, hash calculation, and random number calculation are represented by Ten, Tde, Th1 and Th2, respectively; the verification steps include timestamp judgment and Hash verification represented by Tve1 and Tve2, respectively. In the new scheme, during a session establishment process, entity A needs to generate one request datum, two encryption and decryption times, two random number calculations, two timestamp judgments, and one hash calculation, The time required is 2Tge+2Tde+2Ten+2Th1+2Tve1+Th2; the key distribution server ACU needs one decryption, two encryptions, one sending message generation, two timestamp judgments, and one hash calculation, the time required is Tde+2Ten+Tge+2Tve1+Th2+Tve2; Entity B needs two encryptions, three decryptions, two random number calculations, two data generations, four timestamp judgments, one hash verification, and one hash calculation. The time required is 2Ten+3Tde+2Th1+2Tge+4Tve1+Tve2+Th2. As shown in Table 7, the time required to establish a complete session connection in the new scheme is 6Ten+6Tde+5Tge+4Th1+8Tve1+2Tve2+2Th2, and the time required in the original scheme is 6Ten+6Tde+5Tge+4Th1. Therefore, the total time consumption of the improved scheme is increased compared to that of the original scheme, which is due to the addition of timestamp judgment, hash calculation, and hash verification.

**Table 7.** Comparison of communication time consumption

| Type | Original Scheme | New Scheme |
|---|---|---|
| Entity A | 2Tge+2Tde+2Ten+2Th1 | 2Tge+2Tde+2Ten+2Th1+2Tve1+Th2 |
| ACU | Tde+2Ten+Tge | 1Tde+2Ten+Tge+2Tve1+Th2+Tve2 |
| Entity B | 2Ten+3Tde+2Th1+2Tge | 2Ten+3Tde+2Th1+2Tge+4Tve1+Tve2+Th2 |
| Total time | 6Ten+6Tde+5Tge+4Th1 | 6Ten+6Tde+5Tge+4Th1+8Tve1+2Tve2+2Th2 |

*6.2. Security Analysis*

TAR-ATK: Tampering attack. The attacker intercepts the sender's plaintext message and replays it to the receiver after being tampered with. In the improved EIBsec protocol, the sender uses the parameters in the message to calculate the hash when communicating with the ACU, even if the attacker intercepts and tampers with the hash in the message. When the ACU and the receiver receive the message, the hash will be recalculated and compared. If the hash values are found to be different, the message will be discarded. The one-way nature of Hash [37,38] solves tampering attacks and security vulnerabilities very well.

REY-ATK: Replay attack. The attack method is that the attacker repeatedly sends an already-received message to the receiver to occupy or consume system resources. In the improved EIBsec protocol, we use the method of adding timestamps to defend against replay attacks. When entity A, ACU, and entity B communicate with each other as senders or receivers, they will all bring timestamps in the messages [39]. After receiving the message, first, it will be verified whether the timestamp exceeds the upper limit of time consumed by normal message sending. If the timestamp is judged to have timed out, the message will be discarded.

SPF-ATK: Spoofing attack refers to an attacker using a fake or disguised identity to communicate with other legitimate hosts or send false messages, causing errors to occur on the host under attack. Both the original protocol and the improved protocol contain random numbers for identity authentication, whether in the session key distribution phase or the mutual identity authentication phase, and new random numbers will be generated each time during the session interaction. The receiver recalculates the random number through the calculation rules agreed in advance to ensure that it will not be spoofed.

Malicious instruction(MI): Malicious instruction refers to an attacker injecting malicious commands into messages to disrupt the normal operation of the system or make the target host a puppet machine or springboard that he can operate. In the improved EIBsec protocol, the attacker cannot obtain a valid session key. Even if the attacker injects malicious instructions into the message, the receiver will verify the timestamp and hash in the message. Therefore, the attacker's message with malicious instructions cannot pass verification and will not cause damage to the system and data.

Forward Security (FS): Forward security means that after the current key is obtained by an attacker, the historical keys are still safe. In the new scheme, each communicating party will generate a random number and perform hash verification and timestamp judgment every time a session connection is established. The random number used to calculate the node key or group key is credible. This ensures that leakage of current session keys does not affect historical communication messages.

Backward Security (BS): Backward security means that after the current key is obtained by an attacker, future keys are still safe. In the new scheme, it is consistent with forward security, except that the generation of keys is consistent with the freshness of random numbers. The attacker cannot predict in advance the session key and node key used in the next session communication, and the receiver will verify the freshness of the message by judging the timestamp, ensuring that the leakage of the current session key will not affect future communication messages.

Through the above analysis, Table 8 compares the security of the original and the improved protocol. it is fully demonstrated that the improved solution provides higher security.

**Table 8.** Security comparison of the original protocol and improved protocol.

| Protocol | TAR-ATK | REY-ATK | SPF-ATK | MI | FS | BS |
|---|---|---|---|---|---|---|
| EIBsec | × | × | ✓ | × | × | × |
| Improved EIBsec | ✓ | ✓ | ✓ | ✓ | ✓ | ✓ |

✓ indicates that it can resist attacks or has this security attribute. × indicates that it cannot resist attacks or does not have this security attribute.

## 7. Conclusions

This article focuses on the security of the EIBsec protocol. A method combining CPN theory and the Dolev–Yao attack model was used to evaluate the security of the protocol. First, we use CPN Tools to model the protocol and verify the consistency of the model with the original protocol based on the state space report. The purpose of this step was to achieve better improvements and security assessments on the model to improve the scheme. Subsequently, in order to evaluate the security of the original protocol, we introduced the Dolev–Yao attacker model into the original model we built. After analysis and evaluation, we found that the original protocol had Security vulnerabilities in tampering and replay attacks. Secondly, we propose improvements to address existing security flaws, modeling the improved protocol through CPN Tools, and adding the same Dolev–Yao attacker model to detect the security of the new scheme. Finally, comparing the performance and security of the original protocol and the improved protocol, it was found that the improved scheme can enhance the security attributes of the protocol, effectively resist man-in-the-middle attacks, and ensure the forward security and backward security of the key [40,41] while being able to resist malicious instruction injection attacks. The improvement scheme proposed in this article is not only applicable to the EIBsec protocol but also has great reference value for other protocols in building automation systems such as BACnet, LonWroks, and OPC-UA [42].

In future work, we plan to use the CPN Tools tool combined with the adversary model to implement more types of attacks with CPN theory to verify whether there are other security vulnerabilities in the protocol. Secondly, by studying more reasonable key agreement [43] and key distribution schemes, we will attempt to design a security protocol that is more suitable for protecting data during the communication process, thereby improving security while reducing communication costs. Finally, a user database will be established in the building automation system, and different access permissions will be set for different users. Risks will be reduced through permission division and a secure authentication mechanism so that ordinary users can share non-sensitive data. And we will synchronous and asynchronous processing mechanisms to the system, count the number of online users in the building automation system, and stagger the peak period of using the system to improve the utilization rate of the system.

**Author Contributions:** T.F. participated in the feasibility discussion, analysis of the paper scheme, the proofreading of the paper, and overall control of paper quality; B.Z. responsible for overall design, protocol CPN modeling, performance analysis, and paper writing. All authors have read and agreed to the published version of the manuscript.

**Funding:** This work is supported by the National Natural Science Foundation of China (Grant No. 62162039, 61762060).

**Acknowledgments:** We acknowledge Feng's guidance on our academic research methods and writing.

**Conflicts of Interest:** The authors declare no conflict of interest.

## Abbreviations

The following abbreviations are used in this manuscript:

| | |
|---|---|
| EIB | European Installation Bus. |
| CPN | Colored Petri Net. |
| BAS | Building Automation Systems. |
| IoT | Internet of Things. |
| TAR-ATK | Tampering attack. |
| REY-ATK | Replaying attack. |
| SPF-ATK | Spoofing attack. |
| MI | Malicious instruction. |
| FS | Forward Security. |
| BS | Backward Security. |

## Appendix A. System Configuration

| | |
|---|---|
| OS: | Windows 11 Professional Edition version 21H2, 64-bit operating system |
| Processor: | Intel(R) Core(TM) i7-12700H 2.30 GHz |
| Software: | CPN Tools version 4.0.1 |
| jdk: | version "20.0.2" |

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
