# Peer review of "Security Evaluation and Improvement of the Extended Protocol EIBsec for KNX/EIB"

_information, doi:10.3390/info14120653_

Round 1

Reviewer 1 Report

Comments and Suggestions for Authors

This paper claims to introduce a Security Evaluation and Improvement of the Extended Protocol

EIBsec for KNX/EIB.

The paper reflects a relatively substantial work; however, there are significant issues in the proposed method. For more details, please refer to my comments on your submitted paper.

Below are some of my primary concerns:

·        The authors should clearly define:

o    Their input compared to the existing methods.

o   The applicability of their proposed method

o   They use the mathematical modeling language Colored Petri Net (CPN) Model without a mathematical section explaining how they modified the existing method.

To validate their proposed method, the authors must provide the steps and the environment that helped them with their graphs. They need to reorganize their experiments section to elaborate on their PoC.

·        The authors should explain:

o   How did they obtain the values presented in their tables and graphs?

o   How the obtained results helped them jump to their conclusions.

Comments on the Quality of English Language

 Extensive editing of the English language is required.

Author Response

The authors should clearly define:

1.  Their input compared to the existing methods.

Response: Thank you for pointing this out. We agree with this comment. But, currently, there are almost no papers related to the security research of the EIBsec protocol, and there are also very few papers that propose improvement solutions for the security of the EIBsec protocol. so I am very sorry, the solutions in this article cannot be compared. If some scholars publish security research on the EIBsec protocol in the future, the author of this article will continue to pay attention and compare it with the solution given in this article to improve the solution given in this article.

2.  The applicability of their proposed method

       Response: Thank you for pointing this out. We agree with this comment. Therefore, We have added a description of the applicability of the scheme proposed in this article in Section 7. (page 25)

3. They use the mathematical modeling language Colored Petri Net (CPN) Model without a mathematical section explaining how they modified the existing method.

Response: Thank you for pointing this out. We agree with this comment. Therefore, We added the mathematical expression of CPN which modified the original protocol in Section 5.2. (page 18)

4.  To validate their proposed method, the authors must provide the steps and the environment that helped them with their graphs. They need to reorganize their experiments section to elaborate on their PoC.

Response: Thank you for pointing this out. We agree with this comment. Therefore, we have added the process of using CPN Tools to draw a simple CPN model in Section 2.2, and supplemented the system environment configuration of the CPN Tools modeling process in Appendix A. Regarding POC, the author only implemented the simulation of three types of attacks in this paper, namely tampering attacks, replay attacks and spoofing attacks, so there is no more complex POC. The POC involved in the paper is described in detail in the security assessment model in Sections 4.2 and 5.3. The author has made corresponding modifications to your comments in the manuscript. (page 15 and page 22)

The authors should explain:

1.  How did they obtain the values presented in their tables and graphs?

Response: Thank you for pointing this out. We agree with this comment. Therefore, We have added the simple use of CPN Tools in Section 2.2. The data in the following tables are from the corresponding CPN model, and the model is built using CPN Tools.

2. How the obtained results helped them jump to their conclusions.

Response: Thank you for pointing this out. We agree with this comment. Therefore, after obtaining the experimental data, we draw corresponding conclusions based on Petri net theory. In Section 2.2, Petri net theory is also briefly introduced. (page 4 and page 5)

Reviewer 2 Report

Comments and Suggestions for Authors

Security Evaluation and Improvement of the Extended Protocol EIBsec for KNX/EIB

The novel contribution of the paper is the use of formal analysis methods, specifically Coloured Petri Net theory and the Dolev-Yao attack model, to identify and address security vulnerabilities in the EIBsec protocol.

While the paper provides a thorough analysis of the EIBsec protocol and proposes an improved version, there are a few ways in which it could be improved.

Firstly, the paper could benefit from a more detailed discussion of the limitations and potential drawbacks of the proposed improvements. This could include a more in-depth analysis of the impact on performance and scalability, as well as potential trade-offs between security and usability.  the study focuses solely on the EIBsec protocol and does not address security issues in other building automation systems. Additionally, the proposed improvements to the protocol may require additional computational resources, which could impact performance.

Secondly, the paper could be strengthened by including a more comprehensive evaluation of the proposed improvements. This could involve testing the improved protocol in a real-world setting or conducting additional simulations to validate the results.

Finally, the paper could benefit from a more detailed discussion of the broader implications of the study. This could include a discussion of how the findings could be applied to other building automation systems or how the formal analysis methods used in the study could be applied to other security-critical systems.

The whole document needs proofreading for issues such as missing spaces. There are other typos spread though and formatting inconsistencies present.

Images should be formatted in a similar size and ensure readability without needing to zoom.

Language is generally good with no major issues identified.

References need attention wrt consistency and formatting.  Style changes in places, this could be artefacts from differing authors/documents being merged.

The document as provided is not in the typical format for review in that document is missing line number. This makes it difficult to refer to specifics. Where issues have been found as much context as possible has been provided to locate the item for addressing the issue.

Page 1

Abstract - proofreading needed.

Keywords: EIBsec rather than EIBsec Data Protection Protocol?

Introduction:

Systems(BAS), KNX/EIB[5] - this issue appears elsewhere in the paper. Entire paper needs review for issue like this.

BACnet/IP [6,7] - are both the authors prior works needed to be cited. neither is referred to again. Would work such as Hollinger (2004) or others are not better suited, as it defines the protocol.

AES [35] - should this not be numbered [8] ? other references largely appear in sequence. This is an outlier to the remainder of the paper.

Page 2

"iconic landmarks like the Oriental Pearl Tower in Shanghai" citation needed. This is not covered in reference 27.

"sensors [33], Convert” capitalisation

"Internet of Things gateway" IoT  as the acronym is defined previously ?

"CPN Tools [2]" CPN is not defined in the body text at this point.

Page 3

2. Related work

Reference [13], Reference [14],References [15,16] etc. this is a change in style and awkward. Look at rephrasing.

"framework omnet++ [17]" --> OMNeT++

"the security vulnerabilities [2] of this protocol." no reference is made to EIBsec at all in [2]

Page 4

"services.ACU" spacing

"ACU mainly consists of two parts: a coupling unit and a key service unit. Coupling unit: Implement standard coupler function."   Consider using bullets and removing repetition.

"The protocol uses the AES-128 [19]" uncertain why 19 is needed when the core AES document is used earlier.

Page 5

Fig 2 - centre this in column?

Bold steps to make them stand out in text.

Page 7

Fig 3 - centre this in column? (apply to figures following as well)

Page 8

Fig 4/5/6 are quite complex. the reader needs to be talked though these, or some other clear indication of flow given Fig 7 reads much better regarding this.

Page 10

Unclear what Name column is doing and why / is used ?

Page 11

the NodeDescirptor() - typo

Page 16

extra white space at bottom of page

Page 17

Inconsistent formatting in table 5 - lack of bold.  Numbers would be more readable if right aligned

Page 18

Typo in Table 6, right align values for improved readability. Is it worth adding the delta/ either as a raw value or percentage? This saves the reader having to work this out themselves.

Page 19

Hash [[20,21] solves - this look like manual citations rather than relying on automated methods (or an artefact from earlier versions?)

Malicious instruction(MI): ßspacing again

Table 8 - put the footnotes on separate lines. pacing needs attention

Acknowledgements:

“We acknowledge Teacher Feng’s guidance on ….and writing.”  Unclear if this is a different Feng to the lead author?

References

Consistency with et al. General need for proof reading

Adhere to journal style guide.

1 - pages?

2 - Inconsistent style   first last vs others which are last, initial, missing volume and number for journal

7 - journal name abbreviation is inconsistent with other references

10 - citeseer is not a proper point of reference or publisher. it happens to be where the paper was found.

11 - Repeated information

15 why title in quotation marks. inconsistent with others. Aug.2008 <- this needs checking and correct formatting

16 trailing pp?

18 - Where/how published

20 - inconsistently formatted to other journals. Duplication/extra info : 2022. 2022 ?

25 - space needed onCryptography

29 journal abbreviation

35 - cite institutional author correctly. Check spacing.

Comments on the Quality of English Language

See author notes.

Author Response

Thank you very much for taking the time to review this manuscript. Please find the detailed Responses below and the corresponding revisions/corrections highlighted/in track changes in the re-submitted files

Page 1

Abstract - proofreading needed.

Response: Thank you for pointing this out. We agree with this comment. Therefore, We have proofread the abstract.

Keywords: EIBsec rather than EIBsec Data Protection Protocol?

Response: Thank you for pointing this out. it is EIBsec Protocol, We have revised (page 1)

Introduction:

Systems(BAS), KNX/EIB[5] - this issue appears elsewhere in the paper. Entire paper needs review for issue like this.

Response: Thank you for pointing this out. BAS is the abbreviation for Building Automation Systems. This has been modified to BAS. KNX/EIB[5] The order of reference numbers has been corrected. (page 1)

BACnet/IP [6,7] - are both the authors prior works needed to be cited. neither is referred to again. Would work such as Hollinger (2004) or others are not better suited, as it defines the protocol.

Response: Thank you for pointing this out. The examples and references [4,5] in the revised manuscript are cited here to illustrate that BACnet/IP, as a protocol widely used by BAS, has authentication flaws. Similarly, EIBsec for KNX/EIB may also have the same authentication flaws. (page 1)

AES [35] - should this not be numbered [8] ? other references largely appear in sequence. This is an outlier to the remainder of the paper.

Response: Thank you for pointing this out. We agree with this comment, Therefore, We have numbered the references sequentially. (page 1)

Page 2  

"iconic landmarks like the Oriental Pearl Tower in Shanghai" citation needed. This is not covered in reference 27.

Response: Thank you for pointing this out. We agree with this comment. Therefore, We have added relevant references to references [9,10] in the revised manuscript. (page 2)

"sensors [33], Convert” capitalization

Response: Thank you for pointing this out. we have modified (page 2)

"Internet of Things gateway" IoT  as the acronym is defined previously ?

Response: yes, it is an acronym

"CPN Tools [2]" CPN is not defined in the body text at this point.

Response: Thank you for pointing this out.  CPN Tools is only a tool, We provide the corresponding understanding in Sections 2.1 and 2.2.

Page 3

"framework omnet++ [17]" --> OMNeT++

Response: Thank you for pointing this out. We agree with this comment, Therefore, References [25] are in the revised manuscript. (page 3)

"the security vulnerabilities [2] of this protocol." no reference is made to EIBsec at all in [2]  

Response: Thank you for pointing this out, we have modified. Therefore, References[2,23] are in the revised manuscript. (page 3)

Page 4

"ACU mainly consists of two parts: a coupling unit and a key service unit. Coupling unit: Implement standard coupler function."   Consider using bullets and removing repetition.

Response: Thank you for pointing this out. We agree with this comment, We used bullet points and eliminated duplication. (page 6)

"The protocol uses the AES-128 [19]" uncertain why 19 is needed when the core AES document is used earlier.

Response: Thank you for pointing this out,We agree with this comment. We feel that AES-128 [19] is duplicated here and the previous reference, so references 19 and 35 are merged in the same place. References[7,8] in the revised manuscript. (page 7)

Page 5 

Fig 2  - centre this in column?  Bold steps to make them stand out in text.

Response: Thank you for pointing this out, We agree with this comment. Therefore, We centered the Fig5. Fig2 corresponds to Fig5 in the revised manuscript (page 7)

Page 7

Fig 3 - centre this in column? (apply to figures following as well)

Response: Thank you for pointing this out, We agree with this comment, We centered the image. Fig 3 corresponds to Fig6 in the revised manuscript  (page 10)

Page 8

Fig 4/5/6 are quite complex. the reader needs to be talked though these, or some other clear indication of flow given Fig 7 reads much better regarding this.

Response: Thank you for pointing this out, We agree with this comment. Therefore, The pictures in this paper are all generated through CPN Tools. If readers want to talk these, readers must master the basic use of CPN Tools. Therefore, we introduce the basic usage of CPN Tools through an example in Section 2.2.  Fig4/5/6 corresponds to Fig7/8/9 in the revised manuscript (page 11/ 12/) 

Page 10  

Unclear what Name column is doing and why / is used ?

Response: The name column is used to represent the name of the node. [480] is the name of a node. (page 13)

Page 11

the NodeDescirptor() – typo

Response: Thank you for pointing this out, The typo has been corrected from NodeDescirptor() to NodeDescriptor() in the revised manuscript. (page 15)

Page 16

extra white space at bottom of page

Response: Thank you for pointing this out, We agree with this comment. Therefore, Additional white space has been adjusted in the revised manuscript.

Page 17

Inconsistent formatting in Table 5 - lack of bold.  Numbers would be more readable if right aligned

Response: Thank you for pointing this out, We agree with this comment. Therefore, We have adjusted the fonts in Table 5 to be consistent. After aligning the numbers to the right, we found that there was a validation error in the typesetting, so the numbers were still displayed in the center. (page 22)

Page 18  

Typo in Table 6, right align values for improved readability. Is it worth adding the delta/ either as a raw value or percentage? This saves the reader having to work this out themselves.

Response: Thank you for pointing this out, We agree with this comment. Therefore, We have corrected the grammatical errors and found a typographical validation error when we right-aligned the numbers, so the numbers are still centered The data in Table 6 was generated through CPN Tools. To Compare the original data and the new plan data. So the author believes that there is no need for delta/ either as a raw value or percentage.  (page 22)

Page 19 

Hash [[20,21] solves - this look like manual citations rather than relying on automated methods (or an artefact from earlier versions?)

Response: Thank you for pointing this out, We agree with this comment, Hash [[20,21] solves, The extra half-square bracket on the left here is due to accidentally inputting it during latex typesetting. References [38,39] in the revised manuscript. (page 23)

Malicious instruction(MI): ßspacing again <corresponds to Page 23 in the revised manuscript>

Response: Thank you for pointing this out, it is on page 23 of the revised manuscript. There is no change in the latex template after the author spaces again.

Table 8 - put the footnotes on separate lines. pacing needs attention

Response: Thank you for pointing this out, Table 8 is on page 24 of the revised manuscript.

Acknowledgements:

“We acknowledge Teacher Feng’s guidance on ….and writing.”  Unclear if this is a different Feng to the lead author?

Response: he is the same person, Tao Feng is my supervisor and one of the contributors to this paper.

References

1 - pages?

Response: Thank you for pointing this out, The revised reference is [1] in the revised manuscript file.

2 - Inconsistent style   first last vs others which are last, initial, missing volume and number for journal

Response: Thank you for pointing this out, The revised reference is [15] in the revised manuscript file.

7 - journal name abbreviation is inconsistent with other references

Response: Thank you for pointing this out, The revised reference is [5] in the revised manuscript file.

10 - citeseer is not a proper point of reference or publisher. it happens to be where the paper was found.

Response: Thank you for pointing this out, We agree with this comment, The revised reference is [18] in the revised manuscript file.

11 - Repeated information

Response: Thank you for pointing this out, The revised reference is [19] in the revised manuscript file. The reference here comes from the Spring official website

15 why title in quotation marks. inconsistent with others. Aug.2008 <- this needs checking and correct formatting

Response: Thank you for pointing this out, We agree with this comment. The revised reference is [23] in the revised manuscript file.

16 trailing pp?

Response: Thank you for pointing this out, Thank you for pointing this out, I am sorry, Author error resulted in an incorrect citation. The revised reference is [24] in the revised manuscript file.

18 - Where/how published

Response: Thank you for pointing this out, I am sorry, An error occurred due to the author's mistake. The revised reference is [26] in the revised manuscript file.

20 - inconsistently formatted to other journals. Duplication/extra info : 2022. 2022 ?

Response: Thank you for pointing this out, The revised reference is [39] in the revised manuscript file.

25 - space needed onCryptography

Response: Thank you for pointing this out, I am sorry, An error occurred due to the author's mistake. The revised reference is [44] in the revised manuscript file.

29 journal abbreviation

Response: Thank you for pointing this out, We agree with this comment, The revised reference is [36] in the revised manuscript file.

35 - cite institutional author correctly. Check spacing.

Response: Thank you for pointing this out, We agree with this comment.  According to the citation address given on the journal's official website, the reference has been modified. The revised reference is [7] in the revised manuscript file.

Reviewer 3 Report

Comments and Suggestions for Authors

An interesting and timely paper. Here are my comments.

Page 1: The sequence of the reference sources is not in order (1 is followed by 4 and 5)? Why does reference 35 suddenly appear. The sequence should be in a logical order throughout the paper.

Page 1: what types of attack are being referred to? Can examples be provided.

Page 2: how is the data managed? Who has access to the data control systems? How many different entities are involved?

Page 2: diversified attacks need explanation, and this relates back to the comment in reference to page 2.

The main research question needs to be stated. The question posed is only for interest it would seem. What about defining the research aim and listing at least 2 research objectives. They could underpin the contributions cited.

It may be useful in the introduction to briefly cover the modelling process and explain why it is an appropriate methodological approach. Also, it is important to outline the methodological approach and indicate its strengths.

There are several minor errors in the text. Please do not use “etc” (page 18) because it does not mean anything.

The paper contains several interesting points, and it can be assumed that they have relevance in terms of the design of systems and government regulation. Can insights be provided in terms of policy and future regulation?  Also, regards the skill set of those that design systems, what needs to be considered as regards ensuring that systems perform as expected and that data is kept secure.  

An interesting point that could be addressed in the conclusion, is how users share information and improve the use of systems. Additionally, risk management could be given more coverage.

An interesting and timely paper. Here are my comments.

Page 1: The sequence of the reference sources is not in order (1 is followed by 4 and 5)? Why does reference 35 suddenly appear. The sequence should be in a logical order throughout the paper.

Page 1: what types of attack are being referred to? Can examples be provided.

Page 2: how is the data managed? Who has access to the data control systems? How many different entities are involved?

Page 2: diversified attacks need explanation, and this relates back to the comment in reference to page 2.

The main research question needs to be stated. The question posed is only for interest it would seem. What about defining the research aim and listing at least 2 research objectives. They could underpin the contributions cited.

It may be useful in the introduction to briefly cover the modelling process and explain why it is an appropriate methodological approach. Also, it is important to outline the methodological approach and indicate its strengths.

There are several minor errors in the text. Please do not use “etc” (page 18) because it does not mean anything.

The paper contains several interesting points, and it can be assumed that they have relevance in terms of the design of systems and government regulation. Can insights be provided in terms of policy and future regulation?  Also, regards the skill set of those that design systems, what needs to be considered as regards ensuring that systems perform as expected and that data is kept secure.  

An interesting point that could be addressed in the conclusion, is how users share information and improve the use of systems. Additionally, risk management could be given more coverage.

An interesting and timely paper. Here are my comments.

Page 1: The sequence of the reference sources is not in order (1 is followed by 4 and 5)? Why does reference 35 suddenly appear. The sequence should be in a logical order throughout the paper.

Page 1: what types of attack are being referred to? Can examples be provided.

Page 2: how is the data managed? Who has access to the data control systems? How many different entities are involved?

Page 2: diversified attacks need explanation, and this relates back to the comment in reference to page 2.

The main research question needs to be stated. The question posed is only for interest it would seem. What about defining the research aim and listing at least 2 research objectives. They could underpin the contributions cited.

It may be useful in the introduction to briefly cover the modelling process and explain why it is an appropriate methodological approach. Also, it is important to outline the methodological approach and indicate its strengths.

There are several minor errors in the text. Please do not use “etc” (page 18) because it does not mean anything.

The paper contains several interesting points, and it can be assumed that they have relevance in terms of the design of systems and government regulation. Can insights be provided in terms of policy and future regulation?  Also, regards the skill set of those that design systems, what needs to be considered as regards ensuring that systems perform as expected and that data is kept secure.  

An interesting point that could be addressed in the conclusion, is how users share information and improve the use of systems. Additionally, risk management could be given more coverage.

An interesting and timely paper. Here are my comments.

Page 1: The sequence of the reference sources is not in order (1 is followed by 4 and 5)? Why does reference 35 suddenly appear. The sequence should be in a logical order throughout the paper.

Page 1: what types of attack are being referred to? Can examples be provided.

Page 2: how is the data managed? Who has access to the data control systems? How many different entities are involved?

Page 2: diversified attacks need explanation, and this relates back to the comment in reference to page 2.

The main research question needs to be stated. The question posed is only for interest it would seem. What about defining the research aim and listing at least 2 research objectives. They could underpin the contributions cited.

It may be useful in the introduction to briefly cover the modelling process and explain why it is an appropriate methodological approach. Also, it is important to outline the methodological approach and indicate its strengths.

There are several minor errors in the text. Please do not use “etc” (page 18) because it does not mean anything.

The paper contains several interesting points, and it can be assumed that they have relevance in terms of the design of systems and government regulation. Can insights be provided in terms of policy and future regulation?  Also, regards the skill set of those that design systems, what needs to be considered as regards ensuring that systems perform as expected and that data is kept secure.  

An interesting point that could be addressed in the conclusion, is how users share information and improve the use of systems. Additionally, risk management could be given more coverage.

Comments on the Quality of English Language

Correction needed of some minor errors.

Author Response

Page 1: The sequence of the reference sources is not in order (1 is followed by 4 and 5)? Why does reference 35 suddenly appear. The sequence should be in a logical order throughout the paper.

Response: Thank you for pointing this out. We agree with this comment. Therefore, We have revised the order of references to follow a logical.

Page 1: what types of attacks are being referred to? Can examples be provided.

Response: Thank you for pointing this out. Therefore, We gave three examples of third-party attack types. (page 2)

Page 2: how is the data managed? Who has access to the data control systems? How many different entities are involved?

Response: Thank you for pointing this out. There are three entities in total, the terminal sensor, the IoT gateway, and the EIBsec system. The data generated during this process are managed uniformly by data security engineers. Residential users should only have access rights, data administrators should have access, add and modify permissions and system administrators should have access, modify, add and delete permissions. (page 2)

Page 2: diversified attacks need explanation, and this relates back to the comment in reference to page 2.

Response: Thank you for pointing this out. Network attackers use diverse attack methods, such as Apache Log4j2 (CVE-2021-4101) remote code execution vulnerability exploitation, Shiro deserialization vulnerability exploitation (CVE-2019-12422), social engineering phishing, etc.(page 2)

The main research question needs to be stated. The question posed is only for interest it would seem. What about defining the research aim and listing at least 2 research objectives. They could underpin the contributions cited.

Response: Thank you for pointing this out. This article mainly studies the security vulnerabilities of the EIBsec protocol and then provides improvements to address the existing security vulnerabilities. There are two research goals in total: the first research goal is to find the security vulnerabilities of the original protocol, and The second research goal is to propose appropriate improvement methods based on the security vulnerabilities of the original protocol. There is only one research subject, this article focuses on the EIBsec protocol as its research subject.

It may be useful in the introduction to briefly cover the modelling process and explain why it is an appropriate methodological approach. Also, it is important to outline the methodological approach and indicate its strengths.

Response: Thank you for pointing this out. We agree with this comment. Therefore, We added Tools Comparison in Section 2.1 and explained why CPN Tools is used in this article and the advantages of CPN Tools. The simple use of CPN Tools is added to Section 2.2. Since Sections 2.1 and 2.2 have a lot of content, adding them to the introduction feels a bit heavy, so the author added them to the related work in Section 2. In Section 3.2. Formal Modeling Proces, the modeling process is described in detail, taking the first message nnaa as an example. (page 4 and page 9).

There are several minor errors in the text. Please do not use “etc” (page 18) because it does not mean anything.

Response; Thank you for pointing this out. We agree with this comment. Therefore, We have removed ect. (page 22)

The paper contains several interesting points, and it can be assumed that they have relevance in terms of the design of systems and government regulation. Can insights be provided in terms of policy and future regulation?  Also, regards the skill set of those that design systems, what needs to be considered as regards ensuring that systems perform as expected and that data is kept secure. 

Response: Thank you for pointing this out. The author of this article only focuses on the security research of the protocol and is temporarily unable to provide insights on policy and future regulation. secondly, Those involved in system design need to master cryptography, and network protocol interaction details, and have a deep understanding of authentication and encryption algorithms.

An interesting point that could be addressed in the conclusion, is how users share information and improve the use of systems. Additionally, risk management could be given more coverage.

Response: Thank you for pointing this out. We agree with this comment. Therefore, We include related prospective work in the second paragraph of the conclusion about how users can share information and improve system use. (page 24)

Round 2

Reviewer 1 Report

Comments and Suggestions for Authors

Thank you for addressing most of my concerns.

Comments on the Quality of English Language

A moderate editing of the English language used in this article is required.